# SCALING AGENTS VIA CONTINUAL PRE-TRAINING

**Liangcai Su[1,2,*], Zhen Zhang[2,*], Guangyu Li[2,3,*], Zhuo Chen[2,3,*], Chenxi Wang[2,4,*],
Maojia Song[2], Xinyu Wang[2,#,*]Kuan Li[2], Jialong Wu[2], Xuanzhong Chen[2], Zile Qiao[2],
Zhongwang Zhang[2], Huifeng Yin[2], Shihao Cai[2], Runnan Fang[2,4], Zhengwei Tao[2],
Wenbiao Yin[2], Rui Ye[2], Yong Jiang[2,#], Ningyu Zhang[4], Pengjun Xie[2], Fei Huang[2],
Kai Ye[1], Kewei Tu[3], Chenxiong Qian[1,#], Jingren Zhou[2]**

[1]The University of Hong Kong, [2]Alibaba Tongyi Lab, [3]ShanghaiTech University,
[4]Zhejiang University, *Equal Contribution, # Corresponding Authors
`wangxinyu.nlp@gmail.com, jiangyong.ml@gmail.com,`
`cqian@cs.hku.hk`

## ABSTRACT

Large language models (LLMs) have evolved into agentic systems capable of autonomous tool use and multi-step reasoning for complex problem-solving. However, post-training approaches building upon general-purpose foundation models consistently underperform in agentic tasks, particularly in open-source implementations. We identify the root cause: the absence of robust agentic foundation models forces models during post-training to simultaneously learn diverse agentic behaviors while aligning them to expert demonstrations, thereby creating fundamental optimization tensions. To this end, we are the first to propose incorporating Agentic Continual Pre-Training (**Agentic CPT**) into the deep research agents training pipeline to build powerful agentic foundational models. Based on this approach, we develop a deep research agent model named AgentFounder. We evaluate our AgentFounder-30B on 10 benchmarks and achieve state-of-the-art performance while retains strong tool-use ability, notably **39.9%** on BrowseComp-en, **43.3%** on BrowseComp-zh, and **31.5%** Pass@1 on HLE.

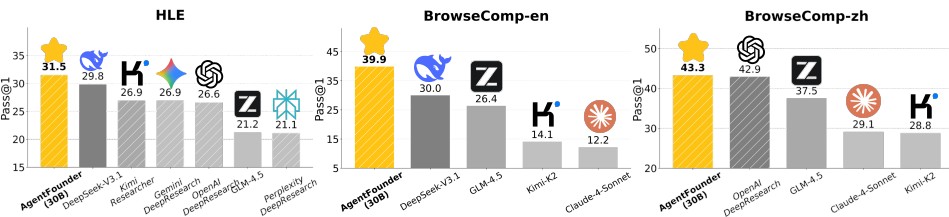

Figure 1: Performance comparison between AgentFounder and state-of-the-art deep research agents.

## 1 INTRODUCTION

The prevailing understanding of *alignment* in language models has been shaped by static interaction paradigms, which focus on ensuring model outputs align with human preferences (Ouyang et al., 2022a). However, as models evolve into autonomous agents capable of multi-step reasoning, dynamic tool invocation, and complex environmental interactions (Yao et al., 2023; Schick et al., 2023; Qin et al., 2024), this definition requires fundamental expansion. To address this paradigm shift, we introduce *agentic alignment*, which requires language models to maintain behavior consistency with human expert demonstrations when solving complex tasks in dynamic environments. This encompasses reasoning chains, tool invocations, and adaptive responses to unpredictable environmental changes (e.g., tool failure, misleading information). Further, language models achieving such alignment for web retrieval and knowledge-intensive tasks can be considered *deep research agents*, capable of autonomously orchestrating sophisticated workflows through search, browsing,

---
*Xinyu Wang leads the project.

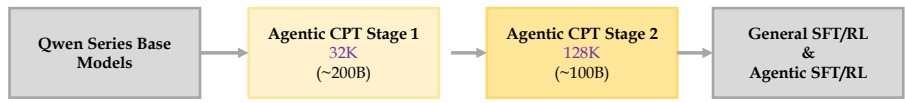

Figure 2: Agentic Training Pipeline.

code execution and other tools to accomplish complex tasks and provide trustworthy answers, e.g., OpenAI Deep Research (OpenAI, 2025b).

Unfortunately, when extended to agentic alignment, current post-training methods, including Supervised Fine-Tuning (SFT) and Reinforcement Learning Fine-Tuning (RL) (Chung et al., 2022; Ouyang et al., 2022a; Taori et al., 2023), demonstrate limited effectiveness, particularly in open-source implementations. Even leading agentic models such as WebSailor (12.0) (Li et al., 2025b), GLM-4.5 (26.4) (Zeng et al., 2025), and DeepSeek-V3.1 (30.0) (DeepSeek-AI, 2025) exhibit substantial performance gaps compared to OpenAI's Deep Research (51.5) on challenging benchmarks like BrowseComp (Wei et al., 2025). A potential explanation is that these methods predominantly rely on general-purpose foundation models[1], such as Qwen2.5-72B (Yang et al., 2024), which presents a critical bottleneck. Specifically, deep research agents require traversing vast policy spaces where SFT's reliance on complete, high-quality trajectory data makes comprehensive coverage infeasible. Moreover, agent trajectories are inherently long and complex, making precise definitions of "correct behavior" challenging. Consequently, both SFT and RL training depend on limited deterministic supervisory signals that lock models into replicating specific behavioral patterns rather than develop flexible decision-making capabilities. Fundamentally, general-purpose foundation models lack agentic inductive biases, forcing post-training to simultaneously learn capabilities and alignment, creating inherent optimization conflicts (Ouyang et al., 2022b; Liu et al., 2021; Haarnoja et al., 2018). Crucially, pathways toward developing agentic foundation models remain largely unexplored.

Thus, beyond post-training, we redefine the agentic alignment training pipeline by introducing Agentic Continual Pre-Training (**Agentic CPT**) as an intermediate scaling layer for agentic alignment. The core objective of Agentic CPT is to deliver a pre-aligned agentic foundation model that naturally supports agentic behaviors for effective downstream fine-tuning. To achieve the goal, agentic CPT operates on two fundamental principles. First, during the data collection phase, seed data sources must be broad and not confined to any single domain. Second, when preparing training data, the training data must comprehensively include various types of agentic behaviors, preventing models from imitating and memorizing specific behavioral patterns that would compromise their behavioral exploration capabilities.

Guided by these principles, we develop **AgentFounder**, a deep research agent model obtained through Agentic CPT and subsequent post-training, starting from Qwen3 series models (Yang et al., 2025). Our agent relies on a systematic and scalable data synthesis approach, comprising First-order Action Synthesis (FAS), Higher-order Action Synthesis (HAS), and a two-stage training strategy.

Then, we evaluate AgentFounder-30B against state-of-the-art (SOTA) models including general LLMs with tools, commercial and open-source deep research agents across 10 benchmarks. AgentFounder-30B achieves superior performance, obtaining **39.9%** on BrowseComp-en, **43.3%** on BrowseComp-zh, **72.8%** on GAIA, **31.5%** on HLE, and **73.0%** on xbench-DeepSearch, achieving new SOTA results. Remarkably, scaling AgentFounder's training data yields steady performance average gains across all benchmarks, exhibiting promising scaling law behaviors. Besides, AgentFounder maintains strong general tool-use capabilities, suggesting potential for general-purpose agents in the future.

## 2 AGENTFOUNDER: AGENTIC FOUNDATION MODEL

This section introduces AgentFounder, our proposed deep research agent that leverages Agentic Continual Pre-Training to create pre-aligned agentic foundation models for downstream fine-tuning.

### 2.1 OVERVIEW OF AGENTIC TRAINING PIPELINE

Standard large language model training pipeline follows a standard two-stage paradigm consisting of pre-training and post-training (*e.g.*, supervised fine-tuning and reinforcement learning). Beyond the

---

[1]GLM-4.5 incorporates synthetic agent trajectories during mid-training.

standard pipeline, we fundamentally redesign the training paradigm by integrating Agentic Continual Pre-Training as an additional stage between pre-training and post-training, as illustrated in Figure 2. During the continual pre-training stage, we still follow the classic next-token prediction paradigm:

$$\mathcal{L} = -\sum_{t=1}^{T} \log P(x_{t+1}|x_1, x_2, ..., x_t) \tag{1}$$

where $P(x_{t+1}|x_1, x_2, ..., x_t) = \text{softmax}(W_o h_t)$, $h_t$ represents the hidden state at position $t$, and $W_o$ is the output projection matrix.

Starting from Qwen's pre-trained foundation models, our enhanced training pipeline consists of:

- Agentic CPT Stage 1: We process approximately 200B tokens of agent data and knowledge reasoning corpora with 32K context length, following the same next-token prediction paradigm as Eq. 1. This stage enables the preliminary acquisition of agentic behaviors including tool invocation patterns and multi-step reasoning chains.
- Agentic CPT Stage 2: We further refine model using 100B tokens of carefully curated, high-quality agent data with extended 128K context windows, allowing the LLM to develop a sophisticated understanding of complex action spaces and long-horizon planning strategies.

## 2.2 First-order Action Synthesis with Zero Supervisory Signal

In this section, we present an agentic data synthesis methodology, First-order Action Synthesis (**FAS**), which operates without supervisory signals and relies solely on diverse data sources.

### 2.2.1 Scaling Training Contexts via Knowledge-to-Question Transformation

Conventional continual pre-training focuses on knowledge adaptation, particularly domain-specific knowledge acquisition (Shi et al., 2024). In contrast, Agentic CPT targets the adaptation of agentic capabilities, which are domain-agnostic abilities that transcend specific domains and enable universal tool utilization and multi-step reasoning. Since these abilities must function effectively across diverse application scenarios, this capability adaptation necessitates training data spanning multiple domains to ensure broad transferability and applicability of the acquired skills.

To achieve this scale and diversity, we collect multiple data types: discarded trajectories from post-training datasets, historical tool invocation results (*e.g.*, search queries and responses), and publicly available corpora (*e.g.*, CommonCrawl (Common Crawl, 2025)). These sources provide continuous scalability while maintaining comprehensive domain coverage. For instance, search results can be persistently extracted from trajectories generated during multiple rollouts in the RL process.

However, while trajectory data directly demonstrates agentic behavior, the remaining sources primarily contain static knowledge. To maximize the utility of this static knowledge for agentic capability training, we reformulate it into diverse training contexts that simulate real-world agent scenarios. This is accomplished through a two-phase transformation approach that systematically generates multi-style questions from static knowledge sources (illustrated in Figure 3):

**Phase 1: Entity-Anchored Open-World Knowledge Memory.** We transform continuously updated unstructured text from various sources into an open-world memory, where entities serve as indexing keys mapping to their associated declarative statements. Unlike traditional knowledge graphs with fixed schemas (Auer et al., 2007; Vrandečić & Krötzsch, 2014), we do not focus on inter-entity relationships, but instead enhance the density of corresponding knowledge statements through reformulation, preserving critical information such as temporal markers, sources, and original stylistic features. For instance, web data containing "The number of tourist arrivals in France increased from 3,793 thousand in May 2025 to 4,222 thousand in June" can be reformulated as: ("France", "Tourist arrivals in France reached 4,222 thousand in June 2025"), rather than limiting to conventional wiki-style knowledge such as "Paris is the capital of France."

**Phase 2: Multi-Style Question Synthesis.** Drawing on the entity-anchored open-world memory, we sample entity clusters along with their associated knowledge statements to synthesize diverse questions spanning factual retrieval, numerical computation, multi-hop reasoning, and synthesis tasks. By exploiting the high density of statements per entity to induce implicit cross-entity links, our

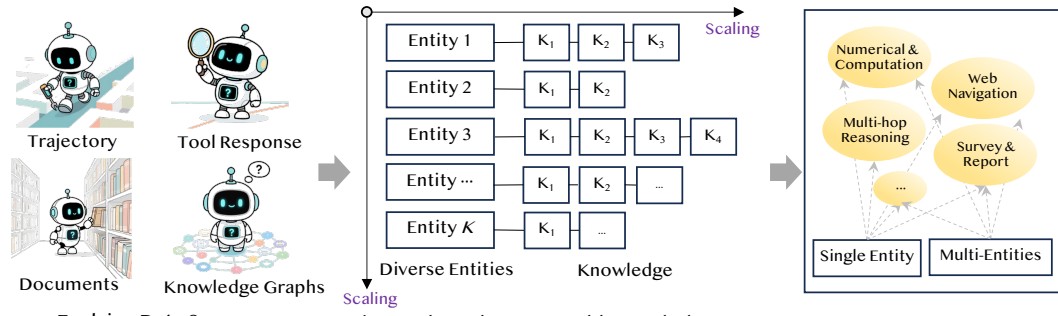

Figure 3: Multi-Style Question-Answer Generation Based on Scalable Data Sources.

approach yields diverse, reliable, and novel questions. Unlike WebSailor (Li et al., 2025b) which requires explicit relationship construction between entities, our method leverages the richness of reformulated knowledge statements from the entity-anchored memory to create natural knowledge intersections, substantially improving the reliability and novelty of question generation. Moreover, the comprehensive knowledge coverage per entity enables synthesis of sophisticated questions even from single-entity contexts. Appendix E.1 provides a generated question-answer example.

### 2.2.2 PLANNING ACTION SYNTHESIS

Beyond large-scale constructed questions, these data alone cannot serve as agent pre-training data due to the lack of corresponding reasoning processes and actions.

**Scalability Challenges.** The intuitive solution is to leverage LLMs to perform tool invocations and reasoning for each question, generating complete agent trajectories. However, this approach faces severe scalability bottlenecks: (1) commercial API costs are prohibitively expensive, particularly for search engine APIs (*e.g.*, Google Search API) and web access APIs (*e.g.*, Jina Reader API (Jina.ai, 2025) and Scraper (scr, 2025)); (2) complete trajectory generation is inefficient and cannot meet the massive data requirements for continual pre-training.

**Insighs.** We observe that the initial analysis of complex problems by LLMs typically involves problem decomposition, information requirement identification, and solution planning, which inherently constitutes high-quality planning data. More importantly, **the quality of first-step reasoning exhibits strong positive correlation with final task completion rates.**

Based on this finding and inspired by multi-reference learning (Zheng et al., 2018; Banerjee & Lavie, 2005), FAS generates multiple reasoning-action data for each problem in planning action synthesis. Given a query $Q$, we employ LLMs to generate $K$ diverse problem analyses along with their corresponding first-step action predictions (tool invocations or direct answers). This approach yields two key advantages: (1) it generates only reasoning chains and tool calls without incurring actual API invocation costs during training data production; (2) the $K$ distinct analytical perspectives effectively expand the action space exploration for each problem.

However, we realize that the aforementioned approach faces two limitations: (1) the generated reasoning-action data may still exhibit similarity despite adjusting parameters like temperature to enhance diversity; (2) it leads to repetitive question text in training data, which is not our optimization target. To address these issues, we propose an improved strategy. Instead of generating $K$ iterations of reasoning-action data for a single question, we generate reasoning-action data for $K$ different questions that share the same knowledge memory but differ in style. This approach better covers training contexts and explores the potential reasoning-action space more comprehensively. In practice, we adopt this question-level diversity expansion as the planning action synthesis method in FAS.

**Reject Sampling with Knowledge Alignment Verification.** Since the planning data may be inaccurate without end-to-end trajectory signals for verification, we implement reject sampling using the accessible knowledge from the question construction process. We employ an LLM-as-Judge approach to evaluate whether the reasoning and actions can effectively acquire the required knowledge, which successfully filters out most low-quality data (detailed in Appendix D.2).

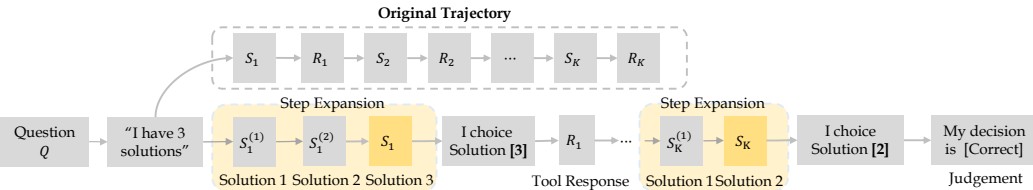

Figure 4: Comparison of high-order action synthesis data and the original trajectory.

### 2.2.3 REASONING ACTION SYNTHESIS

During deep research workflows, a critical scenario emerges when comprehensive information has been acquired through various tool invocations or user interactions: how to effectively synthesize this information to generate final answers or reports. Unlike conventional mathematical reasoning, this reasoning involves logic-based inference grounded in factual information, requiring a balance between divergent and convergent thinking guided by contextual clues, while being difficult to verify through formal methods. Fortunately, questions constructed from knowledge clusters not only provide the questions themselves but also enable precise mapping to the information required for solving them. With this foundation, we propose a two-step logical reasoning data synthesis scheme:

**Step 1:** We require the large language model to decompose the question $Q$ into multiple sub-questions, then leverage its internal knowledge to generate reasonable speculations and answers for each sub-question, producing preliminary answer $A_1$.

**Step 2:** Given the question $Q$ and its mapped requisite knowledge, we require the model to refine answer $A_1$, correct logical errors, and generate the final answer $A_2$.

Notably, we prohibit the model from invoking any external tools during both stages. The motivation behind this two-step design is that if we directly provide both the question and necessary knowledge, the model tends to mechanically utilize the given knowledge as intermediate reasoning nodes rather than simulating an authentic thinking process.

**Reject Sampling.** We employ an LLM-as-judge to evaluate alignment between the generated answer $A_2$ and ground truth, implementing rejection sampling. Correct final answers indicate reliable reasoning processes, enabling us to generate substantial, high-quality logical reasoning chain-of-thought data. This logical reasoning capability is fundamental for agents throughout problem-solving.

### 2.3 HIGH-ORDER ACTION SYNTHESIS WITH SUPERVISORY SIGNAL

**Trajectory-Reuse Challenging.** During the post-training phase of agent models, both reject sampling fine-tuning and RL generate substantial volumes of trajectory data. However, these methods rely heavily on trajectory-level delayed feedback for quality assessment, which results in numerous trajectories being either discarded entirely or utilized only once when they fail to meet stringent quality thresholds. This coarse-grained evaluation approach leads to significant waste of the learning signals embedded within real trajectories. While step-level evaluation could theoretically provide better leverage of these signals, precisely assessing intermediate steps remains challenging. Naively incorporating uncertain supervisory signals risks model collapse, making effective reuse of sub-optimal trajectories while maintaining stability a key challenge.

**Insights.** We recognize that each step in a trajectory is supported by high-quality context including the original question, prior steps, and their real feedback. Each step can actually be viewed as a decision process with different exploration paths, while learning original trajectory data essentially rewards reproducing the full trajectory. We therefore shift the objective from trajectory imitation to step-wise decision-making, explicitly exploiting the choice space at each step to move from trajectory reproduction to decision-based reasoning.

To this end, we introduce **High-order Action Synthesis (HAS)**, a multi-decision action synthesis method through step-level scaling. The core idea is to expand the reasoning-and-action option set at each step to fully explore the local action space, then transform the original trajectory and explored reasoning-action space into decision processing with feedback. Specifically, given a problem $Q$ and a trajectory $T = \{(S_1, R_1), \ldots, (S_K, R_K)\}$, where $S_k$ represents the "reasoning and tool invocation"

at the $k$-th step and $R_k$ denotes the corresponding tool response, the entire trajectory carries a judgment $J \in \{0, 1\}$ indicating failure and success respectively. HAS comprises two components:

**(1) Step-level Scaling.** For any step $S_k$, we denote its conditional context as $C_k = (Q, S_1, R_1, \ldots, S_{k-1}, R_{k-1})$. Without actual tool execution, we employ LLMs to generate $N$ alternative "thought and invocation" candidates for context $C_k$: $A_k = \{S_k^{(1)}, \ldots, S_k^{(N)}\}$. We merge the original step $S_k^{(0)} \equiv S_k$ with these candidates to obtain $N + 1$ feasible steps, then randomly shuffle them to form sequence $\tilde{A}_k$ while recording the original step's position $n_k$ in the sequence. This expansion explores different decision possibilities at the cognitive level, enriching the original trajectory into a decision space with $(N + 1) \times K$ potential reasoning-actions.

**(2) Contrastive Decision-Action Synthesis.** We transform the trajectory with expanded options into a progressive decision-making process. Shown in the Figure 4, starting from problem $Q$ for each step, we explicitly simulate a multi-option selection and decision process. For the $k$-th step, we enumerate each option in $\tilde{A}_k$ and insert an action decision statement: "I will choose option $n_k$", immediately followed by the corresponding real response $R_k$. Finally, we append the judgment text: "My decision is {Correct/Incorrect}" (corresponding to $J$). The complete synthetic training sample is obtained by concatenating the problem, the choice-decision process for each step, and the final judgment text.

This approach avoids uncertain step-level reward risks while enabling learning from diverse reasoning paths, preventing overfitting to specific patterns and transforming underutilized trajectory data into rich training signals.

## 3 EXPERIMENTS

### 3.1 DATA, BASELINES AND BENCHMARKS

**Agentic CPT Data.** Our continual pre-training corpus comprises a carefully curated heterogeneous mixture of: (1) high-quality web-crawled data filtered for factual accuracy, (2) historical tool invocation records, e.g., search results and web page content, (3) offline Wikipedia data, and (4) mixed-quality discarded trajectories from previous post-training iterations.

Based on these data sources, we construct a progressive two-stage curriculum totaling $\sim$300B tokens to systematically enhance the model's agentic capabilities.

**Agentic CPT Stage 1 ($\sim$200B tokens).** This stage focuses on establishing core planning and reasoning skills. The composition includes $\sim$110B tokens of Planning Action data, $\sim$30B of Reasoning Action data, and $\sim$40B of HAS data. To mitigate the catastrophic forgetting of general capabilities, the remaining tokens consist of high-quality general-purpose data, covering mathematics, code, and long-chain reasoning tasks.

**Agentic CPT Stage 2 ($\sim$100B tokens).** To support complex tasks requiring extensive information processing, this stage prioritizes long-context capabilities. It incorporates $\sim$70B tokens of long-context HAS data with sequence lengths ranging from 64K to 128K, alongside a curated set of general-purpose tool-use data. To ensure training stability and distribution consistency, we also replay a $\sim$20B subset (10% downsampled portion) of the data from Stage 1.

**Post-training Data.** To better unlock the potential of our foundation models and demonstrate the adaptability of AgentFounder-Base across different post-training strategies, we employ three distinct SFT configurations named **SFT-A**, **SFT-B** and **SFT-C**. More Details are shown in Appendix C.1.

**Data Decontamination.** To strictly prevent test data leakage, we employ two complementary strategies. For textual overlap, we follow the decontamination protocols of GPT-3 (Brown et al., 2020), PaLM (Chowdhery et al., 2022), and Qwen (Bai et al., 2023) by implementing a strict 13-gram overlap detection to remove training samples that match the test set. Additionally, we conduct URL-based filtering to explicitly exclude domains known to host evaluation benchmarks (e.g., Hugging Face), thereby preventing source-level data leakage.

**Baselines**. We evaluate against three categories of strong agent models:

• **General LLMs with tools**: Qwen3-30B-A3B-2507 (Yang et al., 2025), Qwen3-235B-A22B-2507 (Yang et al., 2025), DeepSeek-R1-0528 (Guo et al., 2025) and Claude-4-Sonnet (Anthropic, 2025).

Table 1: Results on general web search benchmarks. [†] indicates results reported in official sources or prior work.

| Backbone | BrowseComp-en | BrowseComp-zh | GAIA | xbench-DeepSearch | WebWalkerQA |
|---|---|---|---|---|---|
| *General LLMs with tools* | | | | | |
| Qwen3-30B-A3B | 0.5 | 13.5 | 35.9 | 32.0 | 46.9 |
| Qwen3-235B-A22B | 2.3 | 29.4 | 45.6 | 46.0 | 59.6 |
| DeepSeek-R1 | 8.9[†] | 35.7[†] | - | 55.0[†] | - |
| Claude-4-Sonnet | 12.2[†] | 29.1[†] | 68.3[†] | 64.6[†] | 61.7[†] |
| *Commercial Deep Research Agents* | | | | | |
| Kimi-Researcher | - | - | - | 69.0[†] | - |
| OpenAI-o3 | 49.7[†] | 58.1[†] | 70.5[†] | 66.0[†] | 71.7[†] |
| OpenAI Deep Research | 51.5[†] | - | 67.0[†] | - | - |
| *Open-source Deep Research Agents* | | | | | |
| WebThinker-32B-RL | 2.8[†] | 7.3[†] | 48.5[†] | 24.0[†] | 46.5[†] |
| ASearcher-Web-QwQ | 5.2[†] | 15.6[†] | 52.8[†] | 42.1[†] | 34.3[†] |
| WebSailor-72B | 12.0[†] | 30.1[†] | 55.4[†] | 55.0[†] | - |
| WebShaper-72B | - | - | 60.1[†] | - | 52.2[†] |
| AFM-32B-RL | 11.1[†] | - | 55.3[†] | 63.0[†] | - |
| MiroThinker-32B-DPO$_{v0.2}$ | 17.2[†] | 29.4[†] | 64.1[†] | 56.0[†] | 53.6[†] |
| DeepDiver-V2-38B | 13.4[†] | 34.6[†] | - | 53.0[†] | - |
| WebExplorer-8B | 15.7[†] | 32.0[†] | 50.0[†] | 53.7[†] | 62.7[†] |
| DeepDive-32B | 14.8[†] | 25.6[†] | - | 50.5[†] | - |
| Kimi-K2 | 14.1[†] | 28.8[†] | 57.3[†] | 50.0[†] | 63.0[†] |
| GLM-4.5 | 26.4[†] | 37.5 | 66.0[†] | 70.0[†] | 65.6[†] |
| DeepSeek-V3.1 | 30.0[†] | 49.2[†] | 63.1[†] | 71.0[†] | 61.2[†] |
| *Ours* | | | | | |
| **AgentFounder-30B** | **39.9** | **43.3** | **72.8** | **73.0** | **71.9** |

- **Commercial deep research agents**: Kimi-Researcher (Team et al., 2025), OpenAI-o3 (OpenAI, 2025a), OpenAI Deep Research (OpenAI, 2025b), Grok Deeper Search (x.ai, 2025), Perplexity Deep Research (Perplexity AI, 2025), Gemini Deep Research (Google, 2025).

- **Open-source deep research agents**: WebThinker-32B-RL (Li et al., 2025e), ASearcher-Web-QwQ (Gao et al., 2025), WebSailor-72B (Li et al., 2025b), WebShaper-72B (Tao et al., 2025), AFM-32B-RL (Li et al., 2025c), MiroThinker-32B-DPO$_{v0.2}$ (Team, 2025a), DeepDiver-V2-38B (Team, 2025b), WebExplorer-8B (Liu et al., 2025), DeepDive-32B (Lu et al., 2025), Kimi-K2-Instruct (Team et al., 2025), GLM-4.5 (Zeng et al., 2025), and DeepSeek-V3.1 (DeepSeek-AI, 2025).

**Benchmarks**. We evaluate across two categories of agentic benchmarks:

**General web search benchmarks:** *BrowseComp-en* (Wei et al., 2025), *BrowseComp-zh* (Zhou et al., 2025b), *GAIA*[2] (Mialon et al., 2023), *Xbench-DeepSearch* (Xbench-Team, 2025) and *WebWalkerQA* (Wu et al., 2025b), targeting general-purpose browsing, search and reasoning tasks.

**Scenario-targeted web search benchmarks:** *DeepResearch Bench* (Du et al., 2025a), *SEAL-0* (Pham et al., 2025), *Frames* (Krishna et al., 2024), *HLE* (Humanity's Last Exam) (Phan et al., 2025) and *Academic Browse* (Zhou et al., 2025a).

## 3.2 PERFORMANCE COMPARISON BETWEEN AGENTIC MODELS

Table 1 and 2 present a performance comparison of our model AgentFounder-30B under the single-agent ReACT (Yao et al., 2023) paradigm against SOTA models. We have the following observations:

**Observations on General Web Search Benchmarks.** AgentFounder-30B outperforms all existing open-source deep research agents across four benchmarks and achieves comparable performance to DeepSeek-V3.1 on BrowseComp-zh. On BrowseComp-en, it surpasses the best open-source model DeepSeek-V3.1 by 10.0%, closely approaching OpenAI o3 and Deep Research performance, demonstrating effective mastery of sophisticated search strategies and reasoning capabilities. On BrowseComp-zh, AgentFounder-30B's performance (43.3) surpasses GLM-4.5 (37.5) but remains comparable to DeepSeek-V3.1 (49.2) and behind OpenAI-o3 (58.1). We attribute this gap to limited Chinese data in our training corpus and potential Google Search suboptimal performance in Chinese contexts. On the remaining three benchmarks, AgentFounder-30B consistently outperforms all open-source agents and exceeds OpenAI-o3. It achieves the highest single-agent accuracy of 72.8% on GAIA, demonstrating capabilities beyond retrieval reasoning that transfer to broader task categories.

---

[2]We use the text-only subset consisting of 103 questions.

Table 2: Results on Scenario-Targeted Web Search Benchmarks. $^\dagger$ indicates results reported in official sources or prior work.

| Backbone | HLE Pass@1 | DeepResearch Bench RACE Overall | Frames Pass@1 | SEAL-0 Pass@1 | AcademicBrowse Pass@1 |
|---|---|---|---|---|---|
| *General LLMs with tools* | | | | | |
| Qwen3-30B-A3B | 13.2 | 40.2 | 56.4 | 9.9 | 41.3 |
| Qwen3-235B-A22B | 20.0 | 44.8 | - | 14.4 | 50.7 |
| DeepSeek-R1 | 24.8$^\dagger$ | - | 82.0$^\dagger$ | 29.7$^\dagger$ | - |
| Claude-4-Sonnet | 20.3$^\dagger$ | - | 80.7$^\dagger$ | - | - |
| *Commercial Deep Research Agents* | | | | | |
| Grok Deeper Search | - | 38.2$^\dagger$ | - | - | - |
| Perplexity Deep Research | 21.1$^\dagger$ | 40.5$^\dagger$ | - | - | - |
| Gemini Deep Research | 26.9$^\dagger$ | 49.7$^\dagger$ | - | - | - |
| Kimi-Researcher | 26.9$^\dagger$ | 44.6$^\dagger$ | 78.8$^\dagger$ | 36.0$^\dagger$ | - |
| OpenAI-o3 | 20.2$^\dagger$ | - | 84.0$^\dagger$ | - | - |
| OpenAI Deep Research | 26.6$^\dagger$ | 46.5$^\dagger$ | - | - | - |
| *Open-source Deep Research Agents* | | | | | |
| ASearcher-Web-QwQ | 12.5$^\dagger$ | - | 70.9$^\dagger$ | - | - |
| DeepDive-32B | - | - | 76.1$^\dagger$ | 29.3$^\dagger$ | - |
| MiroThinker-32B-DPO$_{v0.2}$ | 17.8$^\dagger$ | - | 74.8$^\dagger$ | - | - |
| WebExplorer-8B | 17.3$^\dagger$ | - | 75.7$^\dagger$ | - | - |
| Kimi-K2 | 18.1$^\dagger$ | 25.4 | 72.0$^\dagger$ | 25.2 | 47.3 |
| GLM-4.5 | 21.2$^\dagger$ | 39.2 | 78.9$^\dagger$ | 34.2 | 55.6 |
| DeepSeek-V3.1 | 29.8$^\dagger$ | 35.4 | 83.7$^\dagger$ | 42.6$^\dagger$ | 65.0 |
| *Ours* | | | | | |
| **AgentFounder-30B** | **31.5** | **47.9** | **89.6** | **43.9** | **75.3** |

Table 3: Adaptability Validation of the AgentFounder Base Model with Different Post-Training Data.

| Base Model | SFT Data | BrowseComp-en | BrowseComp-zh | GAIA | HLE |
|---|---|---|---|---|---|
| Qwen3-30B-A3B-Base | SFT-A | 26.9 | 29.8 | 67.0 | 23.5 |
| AgentFounder-30B-Base | SFT-A | **31.4** | **35.6** | **72.8** | **30.4** |
| *Relative* $\Delta$ | SFT-A | +4.5 | +5.8 | +5.8 | +6.9 |
| Qwen3-30B-A3B-Base | SFT-B | 28.6 | 35.6 | 71.8 | 27.0 |
| AgentFounder-30B-Base | SFT-B | **39.9** | **43.3** | **72.8** | **31.5** |
| *Relative* $\Delta$ | SFT-B | +11.3 | +7.7 | +1.0 | +4.5 |
| Qwen3-30B-A3B-Base | SFT-C | 24.5 | 36.7 | 68.9 | 27.9 |
| AgentFounder-30B-Base | SFT-C | **38.8** | **44.3** | **71.8** | **28.9** |
| *Relative* $\Delta$ | SFT-C | +14.3 | +7.6 | +2.9 | +1.0 |

**Observations on Scenario-Targeted Web Search Benchmarks.** AgentFounder-30B demonstrates impressive performance across specialized tasks. On HLE, it becomes the first open-source model to surpass the 30-point threshold at 31.5%, significantly exceeding all closed-source deep research products including Gemini-2.5-Pro Deep Research, Kimi-Researcher, and OpenAI Deep Research. AgentFounder-30B scores 75.3% on Academic Browse, substantially outperforming existing open-source models as an academic assistant. It outperforms all models on Frames, demonstrating superior multi-perspective reasoning and information synthesis. On Seal-0, it comprehensively outperforms open-source agents, indicating strong resistance to information interference. Finally, it achieves 47.9% on DeepResearch Bench RACE Overall, surpassing OpenAI Deep Research and all open-source agents, confirming comprehensive, readable, and in-depth report generation.

## 3.3 ADAPTABILITY OF THE AGENTIC BASE MODEL TO POST-TRAINING

**Experiment Design.** We adopt the AgentFounder-30B-Base as the pre-alignment agentic base model. We then validate the adaptability of our base model by conducting agentic alignment training with three distinct SFT datasets as described in Appendix C.1: SFT-A, SFT-B, and SFT-C.

**Observation Results.** Table 3 presents the results, from which we make the following observations:

**(1) Agentic CPT demonstrates consistent and substantial improvements.** Specifically, models trained on SFT-A, SFT-B, and SFT-C datasets show average performance gains of 5.75%, 6.13%, and 6.45% respectively when built upon AgentFounder-30B-Base.

**(2) Post-training data remains crucial for unlocking base model capabilities.** Despite sharing the same AgentFounder-30B-Base foundation, models exhibit significant performance variations across different post-training datasets. For instance, on BrowseComp-zh, AgentFounder-30B with SFT-B outperforms variants with SFT-A and SFT-C by 8.5% and 3.0% respectively. This underscores the irreplaceable role of agentic post-training and highlights the ongoing challenge of fully realizing base model potential through optimal training strategies.

## 3.4 IMPACT ON GENERAL CAPABILITIES

**Experiment Design.** To mitigate potential catastrophic forgetting, we explicitly incorporated high-quality general-purpose data into our training mixture. We evaluate the performance on three widely recognized general benchmarks: MMLU Li et al. (2023), SuperGPQA Du et al. (2025b), and GPQA Rein et al. (2023). In this analysis, we compare the original foundation model initialization before any agentic adaptation, denoted as *Qwen3-30B-A3B-Base*, against our agentic base model, *AgentFounder-30B-A3B-Base* (prior to Agentic SFT), to quantify the changes in general ability.

Table 4: Comparison of general capabilities.

| Benchmark | Qwen3-30B-A3B-Base | AgentFounder-30B-Base | $\Delta$ |
|---|---|---|---|
| MMLU | 81.38 | 80.11 | -1.27 |
| SuperGPQA | 35.72 | 34.27 | -1.45 |
| GPQA | 43.94 | 42.58 | -1.36 |

**Results and Analysis.** As presented in Table 4, we observe a marginal regression in performance across the general benchmarks. Specifically, AgentFounder-30B-A3B-Base exhibits a decrease of 1.27 points on MMLU, 1.45 on SuperGPQA, and 1.36 on GPQA compared to the Qwen3-30B-A3B-Base. This slight degradation is primarily attributable to the relatively small proportion of general-purpose data in the mixture compared to the massive volume of agentic data. However, considering the significant gains achieved in agentic capabilities, we deem this minor drop an acceptable trade-off. The results confirm that our strategy of mixing general data effectively mitigates severe catastrophic forgetting, maintaining the model's foundational knowledge at a robust level.

## 3.5 ABLATION STUDIES

### 3.5.1 IMPACT OF TRAINING STRATEGIES

We examine whether our proposed two-stage training paradigm can bring performance improvements.

**Experiment Design.** We set the training tokens to 50B, using Qwen3-30B-A3B-Base as the initialization checkpoint and employing SFT-A data. We compare two configurations: AgentFounder Stage 1, which applies single-stage training on all data where some HAS data may be truncated due to length constraints, and AgentFounder Stage 1&2, which implements our complete two-stage training approach that specifically incorporates synthesized long-context agent data in the second stage.

**Observation Results.** As shown in Table 5, our analysis demonstrates that the two-stage training paradigm yields consistent performance improvements. Specifically, the AgentFounder Stage 1&2 configuration achieves an average improvement of 3.3% on Pass@1 and 3.7% on Pass@3 across the three evaluation tasks compared to the single-stage baseline. These results substantiate the necessity of learning complete long-context agent data rather than truncated sequences. While resource constraints preclude evaluation of single-stage training with extended context lengths (e.g., 128K), such approaches would incur substantially higher computational costs.

### 3.5.2 IMPACT OF DATA TYPES

**Experiment Design.** We investigate the effectiveness of the two data types introduced in this work for agentic CPT. We conduct single-stage training experiments using approximately 50B tokens of FAS data and FAS+HAS mixed data respectively, followed by post-training with SFT-A data.

**Observation Results.** Table 6 shows that both FAS and HAS data contribute meaningful performance improvements: **(1) FAS data demonstrates clear efficacy.** Training with FAS data yields substantial

Table 5: Evaluation of the effectiveness of the two-stage AgentFounder training strategy.

| Strategy | BrowseComp-en | | BrowseComp-zh | | GAIA | |
|---|---|---|---|---|---|---|
| | Pass@1 | Pass@3 | Pass@1 | Pass@3 | Pass@1 | Pass@3 |
| AgentFounder Stage 1 Only | 31.4 | 49.9 | 34.3 | 50.5 | 69.9 | 81.6 |
| AgentFounder Stage 1 & 2 | 35.5 | 52.0 | 37.2 | 58.5 | 72.8 | 82.5 |
| *Relative* Δ | **+4.1** | **+2.1** | **+2.9** | **+8.0** | **+2.9** | **+0.9** |

Table 6: Effect of data type (HAS and FAS).

| Data | Tokens | BrowseComp-en | | BrowseComp-zh | | GAIA | |
|---|---|---|---|---|---|---|---|
| | | Pass@1 | Pass@3 | Pass@1 | Pass@3 | Pass@1 | Pass@3 |
| Non CPT | 0B | 26.9 | 38.0 | 29.8 | 45.3 | 67.0 | 75.7 |
| FAS | 50B | 31.4 | 49.9 | 37.0 | 54.3 | 72.8 | 80.6 |
| FAS+HAS | 50B | 31.4 | 50.1 | 40.1 | 54.7 | 69.9 | 82.5 |
| *Relative* Δ | / | 0.0 | **+0.2** | **+3.1** | **+0.4** | **-2.9** | **+1.9** |

gains, particularly in Pass@3 metrics. On BrowseComp-zh, FAS-based continual pre-training achieves 9.0% improvement, establishing higher performance ceilings for post-training. **(2) HAS provides complementary benefits.** FAS+HAS combination consistently delivers positive gains across benchmarks. While GAIA shows 2.9% Pass@1 decrease, the corresponding 1.9% Pass@3 improvement suggests normal evaluation fluctuation rather than systematic degradation.

## 3.6 SCALING LAW EXPLORATION

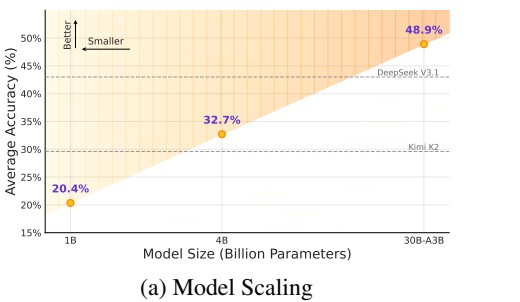

(a) Model Scaling

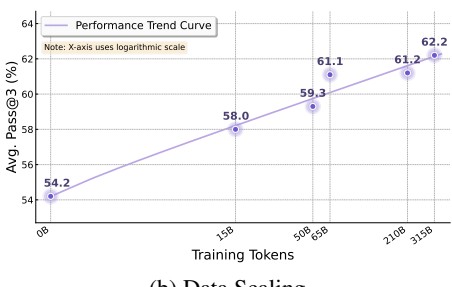

(b) Data Scaling

Figure 5: Scaling Law Exploration for Agentic Capabilities.

**Scaling Law Analysis.** We systematically explore scaling laws across model size and training data volume. Our experiments reveal that agentic capabilities consistently improve with both model size scaling (1B→4B→30BA3B) and data volume scaling (15B→315B tokens). Specifically, data volume scaling improves average accuracy across eight benchmarks from 54.2% to 62.2%, demonstrating effective scaling of agentic CPT data. However, whether the performance gains from model scaling stem from the MoE architecture (with reduced activation) warrants further investigation. More details are provided in Appendix D.1.

## 4 CONCLUSION

In this work, we redefine the training pipeline for agentic alignment in deep research agents by introducing Agentic Continual Pre-Training (Agentic CPT) beyond traditional post-training for the first time. Furthermore, we present a systematic agentic CPT method comprising scalable agentic data synthesis and an adapted two-stage training strategy. Specifically for agentic data synthesis, we propose first-order action synthesis without additional commercial API calls, including planning action synthesis and reasoning action synthesis to enhance the model's planning and logical reasoning capabilities. Additionally, we provide a higher-order action synthesis method that remodels trajectories as multi-step decision-making problems, leveraging step-level expansion to thoroughly explore solution paths. Built on large-scale offline synthesized HAS and FAS data, we develop a powerful agentic model, AgentFounder-30B, which surpasses current closed-source models across 10 benchmarks, establishing a new state-of-the-art.

## 5 ACKNOWLEDGMENT

This work is partially supported by the RGC for Early Career Scheme (No.27210024). Any opinions, findings, or conclusions expressed in this material are those of the authors and do not necessarily reflect the views of RGC.

## 6 ETHICS STATEMENT

We confirm that all training data used in this work consists of legally available and permissible datasets. All authors have strictly adhered to ICLR's ethical guidelines and standards throughout the research process. Our work does not involve or promote content related to bias, violence, privacy violations, or other ethically problematic materials.

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

## A    The Use of Large Language Models (LLMs)

We used LLMs solely for grammar checking and minor typo corrections in this paper. No "prompt injections" are present in the submission. Additionally, given that this paper aims to develop more powerful agentic LLMs, our experimental methodology (e.g., using LLMs as judges) and illustrative examples necessarily involve LLM usage, which does not pose additional academic risks. All instances of LLM use have been clearly disclosed within the paper, and we confirm that our usage fully complies with the stated policy and academic ethics standards.

## B    Related Work

### B.1    Deep Research Agents

Deep research agents are language models capable of autonomous tool use and multi-step reasoning—such as search, web access, code execution, and file parsing—to accomplish research-level tasks and produce reliable reports. OpenAI pioneered this field with a closed-source agent that analyzes hundreds of web pages and generates cited reports within minutes (OpenAI, 2025b), inspiring similar products from Grok (xAI, 2025), Perplexity (Perplexity AI, 2025), Google Gemini (Google, 2025), and Kimi (Moonshot AI, 2025).

**Open-source deep research agents** Open-source deep research agents have achieved strong results on benchmarks such as BrowseComp-en (Wei et al., 2025), BrowseComp-zh (Zhou et al., 2025b), Xbench-DeepSearch (Xbench-Team, 2025), and GAIA (Mialon et al., 2023), with notable systems ranging from early works like Search-o1 (Li et al., 2025d), R1-Searcher (Song et al., 2025), EvolveSearch (Zhang et al., 2025a), MaskSearch (Wu et al., 2025c), and WebThinker (Li et al., 2025e), to later agents tackling harder tasks, such as MiroThinker (Team, 2025a). A common focus is constructing challenging problems and trajectories for training, leading to diverse data-generation strategies: iterative complexity growth (WebDancer (Wu et al., 2025a), ASearcher (Gao et al., 2025)), knowledge graph-based synthesis (WebSailor (Li et al., 2025b)), information-seeking task modeling (WebShaper (Tao et al., 2025)), Chain-of-Agents pipeline (AFM (Li et al., 2025c)), open knowledge graph sampling (DeepDive (Lu et al., 2025)), cross-page/riddle generation (Deepdiver (Team, 2025b)), and autonomous exploration (WebExplorer (Liu et al., 2025)). While these methods enhance agent capability via supervised SFT or RL, they largely overlook continual pre-training. Recent open-source models such as Kimi-K2 (Team et al., 2025), GLM-4.5 (Zeng et al., 2025), and DeepSeek-V3.1 (DeepSeek-AI, 2025) have begun emphasizing agentic capabilities, but systematic exploration of continual pre-training for agents remains limited.

**Multi Deep Research Agents and Multi-modal Deep Research Agents.** Beyond training methodologies, innovative inference paradigms have also emerged. Tencent's contribution extends beyond their trained CK-Pro-8B model to include the open-sourced Cognitive Kernel-Pro multi-agent framework (Fang et al., 2025b). Tencent Youtu has developed a multi-agent framework utilizing DeepSeek-V3.1, achieving 71.47% accuracy on the WebWalkerQA benchmark (Wu et al., 2025b). Additional notable contributions include SkyworkAI's DeepResearchAgent (Zhang et al., 2025b) and ByteDance's deer-flow (ByteDance, 2025), which further explore the potential of agent models. In the multimodal domain, Alibaba Tongyi Lab has open-sourced WebWatcher (Geng et al., 2025), the first multimodal deep research agent.

### B.2    Continual Pre-training

While LLMs are initially pretrained on large general corpora, research has shown that continual pre-training (CPT) can further boost performance through ongoing unsupervised training (Gupta et al., 2023; Lin et al., 2025; Jin et al., 2022). Ke et al. (2023) introduce continual domain-adaptive pre-training, which mitigates catastrophic forgetting and enables knowledge transfer for better end-task results. Extending this, Çağatay Yıldız et al. (2025) study billion-parameter models, finding CPT consistently improves performance for models under 1.5B parameters and surpasses standard domain adaptation. Parmar et al. (2024) offer guidelines on data distribution and learning-rate scheduling for CPT, showing gains when applied to a 15B-parameter model over its original pre-training baseline. Together, these studies position CPT as a promising method for enhancing LLM capabilities.

However, regarding CPT and nowadays agents development, existing work focuses primarily either on CPT on tasks without tool calling or on post-training (such as SFT and RL) for agent development. Integrating agentic capabilities directly via the continual pre-training phase remains largely unexplored. This motivates our exploration of **Agentic CPT** as a novel paradigm that embeds agentic reasoning and tool-use capabilities at the foundational pre-training level.

## C    EXPERIMENTAL SETUP DETAILS

### C.1    SFT DATA CONFIGURATION

In this paper, we adopt three SFT data configuration as follows:

- **SFT-A:** Employs a two-stage training paradigm, first on general conversational data, followed by specialized React-style agent trajectories with explicit reasoning chains.
- **SFT-B:** An enhanced version of SFT-A that maintains the two-stage training paradigm but incorporates a balanced mixture of general conversational data and React-style trajectories in each stage.
- **SFT-C:** Employs a two-stage training paradigm with general conversational SFT data and React with summarized reasoning trajectories.

For all SFT configurations, the set of the challenging information-seeking questions is constructed following the methodology from WebSailor-V2 (Li et al., 2025a), WebResearcher (Qiao et al., 2025), WebWeaver (Li et al., 2025f) and AgentScaler (Fang et al., 2025a).

### C.2    TOOLS

In this work, we utilize five different tools for our AgentFounder model, namely Search, Visit, Python Interpreter, Google Scholar, and File Parser:

- **Search** leverages the Google search engine for large-scale information retrieval. The tool accepts a list of one or more search queries to be executed concurrently. For each query, it returns the top-10 ranked results, with each result comprising a title, a descriptive snippet, and its corresponding URL.
- **Visit** is designed for targeted information extraction from web pages. The tool takes as input a set of web pages, where each page is paired with a dedicated information-seeking goal. The process begins by employing Jina (Jina.ai, 2025) to retrieve the full content of a given web page. Subsequently, a summary model processes this content to extract only the information pertinent to that page's specific goal.
- **Python Interpreter** is used to execute Python code within a sandboxed environment. The input is a string of Python code, which must be enclosed within  tags for proper execution. The tool runs the provided code and captures its standard output; therefore, any results or values intended to be seen must be explicitly passed to the print() function. This capability enables dynamic computation, data manipulation, and the use of various Python libraries in a secure and isolated manner.
- **Google Scholar** is used to retrieve information from academic publications. The input consists of a list of one or more search queries, allowing for multiple, distinct searches within a single tool call. The tool leverages the Google Scholar search engine to execute each query and gather relevant scholarly literature, such as articles, papers, and citations.
- **File Parser** answers user queries by analyzing a mix of documents, web pages, and multimedia files (e.g., PDF, DOCX, MP4) from local or URL sources. It works in two steps: first, it converts all input into plain text, transcribing audio/video content when necessary. Second, a summary model reads this unified text to generate a direct answer to the user's question.

### C.3    HYPER-PARAMETERS

When evaluating our AgentFounder models, we employ specific inference parameters to ensure stable and reproducible results: temperature 0.85, repetition penalty 1.1, and top-p 0.95. These settings are

recommended based on extensive empirical validation to optimize the balance between creativity and consistency in agentic reasoning tasks. We set a maximum tool usage limit of 128 calls per task and constrain the context length to 128K tokens.

# D  ADDITIONAL EXPERIMENTAL RESULTS

## D.1  SCALING LAW EXPLORATION

### D.1.1  SCALING WITH MODEL SIZE

**Experiment Design**. We investigate how model scale influences agentic capabilities by evaluating models with 1B, 4B, and 30B parameters on deep research benchmarks. Additionally, we compare our models with two larger-scale baseline models (DeepSeek-V3.1 and Kimi-K2) to assess the efficiency of our agentic continual pre-training approach across different model scales.

**Observation Results**. Our experimental results reveal that model scale plays a crucial role in agentic performance, with our approach demonstrating superior scaling efficiency: **(1) Consistent scaling benefits**. Model size shows a strong positive correlation with agentic performance. Average accuracy increases from 20.4% for the 1B model to 32.7% for the 4B model and further to 48.9% for the 30B model, indicating that larger models possess enhanced capacity for complex agentic behaviors such as effective tool use and multi-step reasoning. **(2) Superior scaling efficiency**. The AgentFounder-30B model achieves 48.9% accuracy, exceeding the performance of two larger baseline models, DeepSeek-V3.1 (43.0%) and Kimi-K2 (29.6%), despite their greater size. This suggests that our agentic continual pre-training approach enables more effective utilization of model capacity for agentic tasks. These results demonstrate that combining agentic continual pre-training with appropriate model scaling provides a strong foundation for building high-performance deep research agents.

### D.1.2  SCALING WITH DATA VOLUME

We investigate the scaling properties of Agentic CPT and validate the effectiveness of our two-stage training strategy across varying data volumes.

**Experiment Design**. We train AgentFounder models with data volumes ranging from 0B to 315B tokens, using Qwen3-30B-A3B-Base as the initialization checkpoint. We implement our two-stage training paradigm, where Stage 2 incorporates 128K context window training at 65B and 315B token checkpoints. Performance is evaluated using the average Pass@3 metric across multiple agentic benchmarks to assess scaling behavior and training effectiveness.

**Observation Results**. As shown in the scaling curve, our analysis reveals three key findings:

**(1) Logarithmic scaling law holds for agentic capabilities.** The relationship between training tokens and performance exhibits logarithmic characteristics, with the most substantial improvements (3.8%) occurring within the initial 15B tokens, demonstrating that agentic behaviors can be efficiently acquired through targeted pre-training.

**(2) Two-stage CPT with extended context provides consistent improvements.** Stage 2 CPT with 128K context windows delivers notable gains at both 65B (+1.8% over 50B) and 315B (+1.0% over 210B), indicating that long-context training effectively enhances complex reasoning capabilities even when the base scaling curve approaches saturation.

**(3) Sustained improvements at scale demonstrate robustness.** AgentFounder maintains consistent performance improvements throughout the entire 315B token budget, achieving a total gain of 8.0% (from 54.2% to 62.2%), validating our training methodology's effectiveness in preventing premature convergence.

## D.2  QUALITY ANALYSIS OF FAS DATA ON PLANNING ACTION

**Experiment Design.** We evaluate the quality distribution and filtering effectiveness of FAS planning action data. Starting with FAS synthesized trajectories, we apply a prompt-based weak supervision filter that analyzes the original question, generated trajectory, and metadata to produce binary

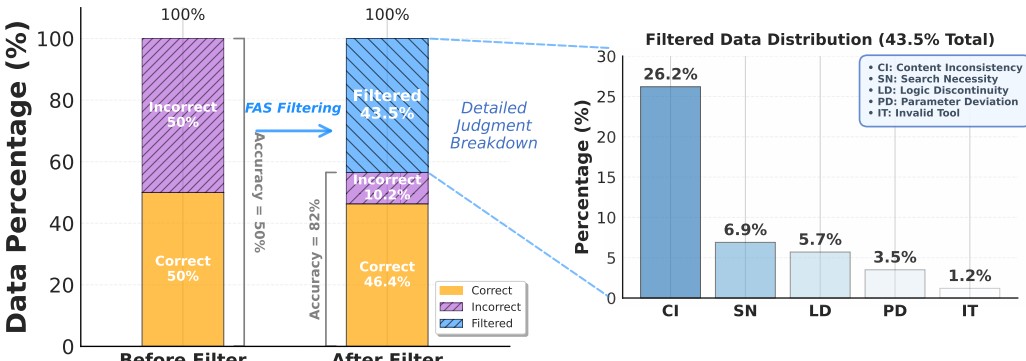

Figure 6: Filtering performance and representative low-quality outputs for weakly supervised filtering in first-order action synthesis.

accept/reject decisions with detailed rejection reasons. We measure the impact on data quality by comparing pre- and post-filtering accuracy rates and analyzing error type distributions.

**Observation Results.** As shown in our analysis, the filtering mechanism significantly enhances FAS planning action data quality: **(1) Filtering achieves substantial quality improvement.** While initial FAS generation yields balanced correct/incorrect trajectories (50%/50%), our filter removes 43.5% of problematic samples, increasing retained trajectory accuracy from 50% to 82%. This confirms FAS effectively generates diverse planning actions requiring quality control for high-fidelity selection. **(2) Semantic errors dominate rejection patterns**. Content Inconsistency accounts for 26.2% of rejections, followed by Search Necessity (6.9%) and Logic Discontinuity (5.7%). The concentration in semantic rather than syntactic errors (Invalid Tool: 1.2%) indicates FAS maintains structural validity while requiring refinement in semantic alignment. **(3) Quality improvement justifies volume reduction.** Though absolute correct data proportion slightly decreases (50% → 46.3%), the 82% accuracy among retained samples validates prioritizing precision over recall for effective agentic CPT.

## D.3 COMPREHENSIVE ANALYSES

### D.3.1 TRAINING PROCESS

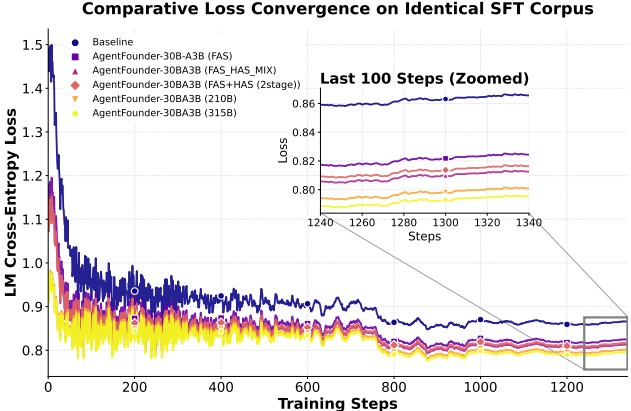

Figure 7: Training loss evolution showing superior convergence of AgentFounder models compared to baseline.

We validate whether agentic continual pre-training can alleviate the dual-burden problem by endowing models with foundational agentic capabilities before post-training.

**Experiment Design.** We conduct a comparative loss analysis between AgentFounder models and the baseline during SFT on identical downstream tasks. All models are trained for 1,340 steps using the same SFT-A data, with training efficiency measured through final loss, minimum achieved loss, and average loss over the last 100 training steps.

**Observation Results.** As shown in Figure 7, our analysis reveals that agentic CPT significantly enhances fine-tuning efficiency: **(1) AgentFounder substantially reduces SFT loss.** All Agent-Founder variants achieve markedly lower loss values compared to the baseline across all metrics. While the baseline reaches a final loss of 0.8656, our best-performing AgentFounder-30B (315B) model achieves 0.7953, demonstrating that agentic CPT effectively endows models with foundational capabilities that facilitate adaptation to downstream agent tasks. **(2) Scaling CPT data yields monotonic improvements.** Loss values decrease progressively as CPT data volume increases from FAS-only to 315B tokens. The FAS+HAS mixture outperforms FAS alone, validating that reorganizing supervision signals from post-training into CPT format strengthens the model's agentic foundation.

### D.3.2 TOOL CALL ANALYSIS

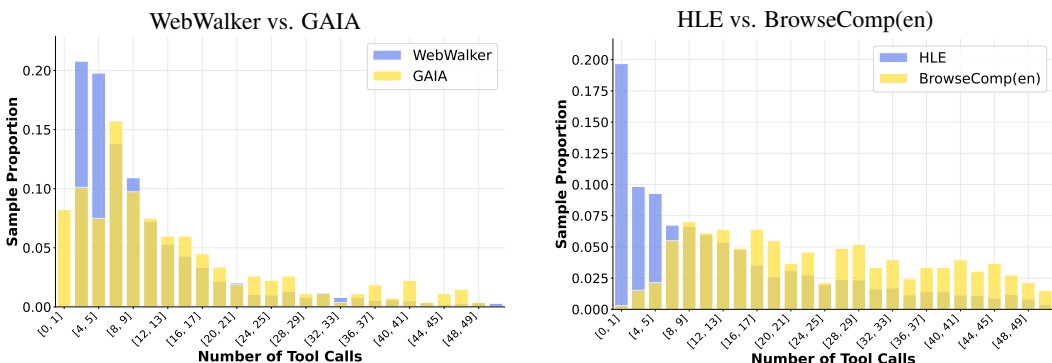

Figure 8: Tool call distribution comparison.

**Experiment Design.** We analyze AgentFounder's tool invocation distributions across four representative benchmarks to understand how the model adapts its tool usage strategies to different task complexities. We examine tool call patterns on HLE, BrowseComp-en, WebWalker, and GAIA-text benchmarks, categorizing them based on their task characteristics and required exploration depth.

**Observation Results.** As illustrated in Figure 8, AgentFounder demonstrates distinct adaptive behaviors across task types: **(1) Complex research tasks exhibit intensive tool usage.** BrowseComp-en and HLE show heavy-tailed distributions with high tool density. BrowseComp-en requires extensive invocations for persistent web browsing, while HLE's extended patterns reflect the need to combine complex reasoning with search-augmented inference. **(2) Structured tasks employ conservative tool usage.** WebWalker's distribution peaks sharply at low invocation counts for efficient text navigation, while GAIA-text exhibits a compact distribution suited for well-defined problems with clear solution paths. These contrasting patterns demonstrate AgentFounder's ability to calibrate tool usage based on task complexity—intensive exploration for open-ended research versus targeted invocations for structured problems.

### D.3.3 GENERAL TOOL-USE ABILITIES

Beyond the tool-use capabilities of Deep Research, we construct agentic CPT data to address more general tool-use scenarios. As shown in Table 7, we compare AgentFounder-30B with Qwen3-30B-A3B on ACEBench (Chen et al., 2025), a benchmark designed to evaluate the tool-use performance of LLMs in diverse, general settings. The results reveal that AgentFounder-30B surpasses Qwen3-30B-A3B in general tool-use ability, suggesting that the Agentic CPT framework can be effectively applied to a broader range of agentic scenarios.

| Model | ACEBench |
|---|---|
| Qwen3-30B-A3B | 67.2 |
| AgentFounder-30B | 70.0 |

Table 7: A comparison of general tool-use ability on ACEBench on overall performance.

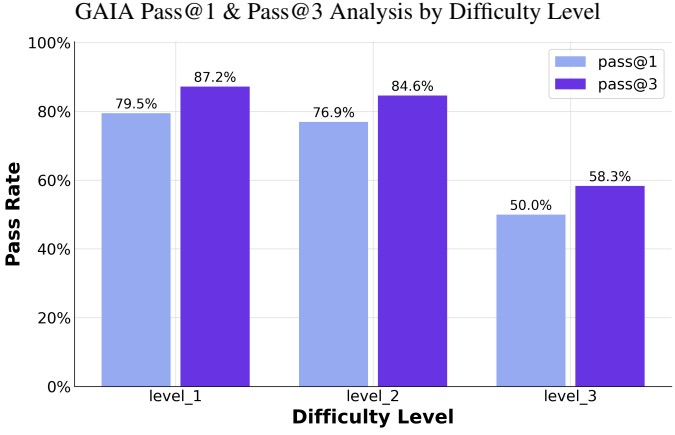

Figure 9: Pass rate on different levels of the GAIA dataset.

## D.4 PERFORMANCE OF GAIA ON DIFFERENT LEVELS

We show the Pass Rate of the GAIA dataset w.r.t. different levels in Figure 9. Specifically, the model achieves its highest performance on level 1 tasks, with a 79.5% Pass@1 rate and an 87.2% Pass@3 rate. A more substantial performance degradation is evident at level 3, where the Pass@1 rate drops to 50.0% and the Pass@3 rate falls to 58.3%. This trend indicates that the model's efficacy is significantly impacted by the complexity of the tasks.

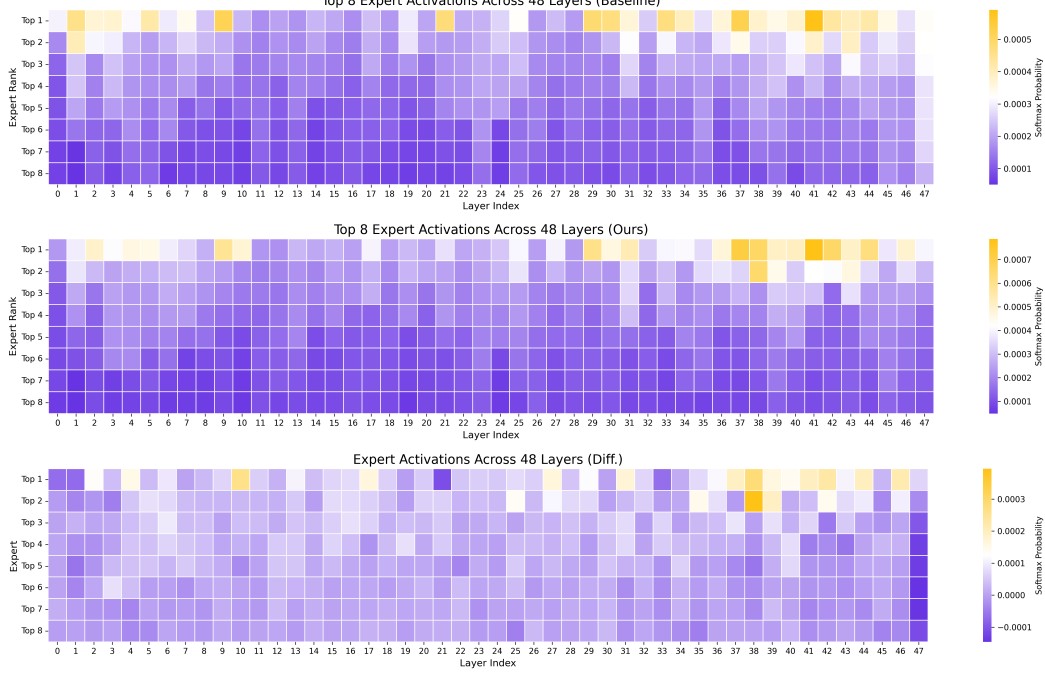

Figure 10: MoE activations illustration on BrowseComp-zh dataset. Top-Middle-Bottom: Baseline, Our model, Difference.

## D.5 MoE ACTIVATIONS

We collect the model's router logits for the questions' last token and display the top 8 ones after softmax. The scores are sorted and averaged over all samples in BrowseComp-zh. The top one shows the activations of the baseline model without applying CPT. The middle one shows the result of our model AgentFounder-30B-A3B. The bottom one shows the difference after calculating ours minus the baseline. It can be observed that, after undergoing CPT, the expert distribution within the MoE architecture becomes more balanced in the final layers of the model, rather than being concentrated. This potentially offers the benefit of enabling more diversified utilization of multiple experts, which empirically leads to greater training stability during the post-training phase. It also helps mitigate the risk of overfitting by preventing the phenomenon of "dead experts"—experts that cease to be activated and thus fail to contribute meaningfully to the model's performance.

## E EXAMPLES

### E.1 GENERATED QUESTION EXAMPLE

---

**Example: Generated Questions with the entity "Paris"**

**Source Entity:** Paris
**Knowledge 1:** *The Louvre welcomed 8.7 million visitors in 2024. Visitor numbers thus remained at 2023 levels (8.9 million visitors) in the rather unique context of the Paris 2024 Olympic and Paralympic Games*
**Knowledge 2:** *In 2023, France's bedbug outbreak sparked a political row; Paris pushed pre-2024 Olympic action, and the transport minister summoned transit operators.*
**Knowledge 3:** *At Paris 2025, Airbus announced 132 firm orders and up to 106 additional options/increases—AviLease (40+37), Riyadh Air (25+25), ANA (27), LOT (40+44)—underscoring Paris as a global aviation deal hub.*
...
**Question:** At the biennial aerospace marketplace named after the city whose pyramid-fronted museum recorded high single-digit millions of visitors during a period of global athletic celebration, and where the year before a citywide nuisance led authorities to convene transit operators, which buyer placed a perfectly balanced commitment with firm orders equal to options?
**Answer:** Riyadh Air

---

This example synthesizes a question from the entity Paris using three news-sourced statements. Knowledge 1 offers an indirect locational anchor via visitor counts at the pyramid-fronted museum during a global sporting period, which localizes the marketplace without naming the city. Knowledge 2 adds a prior-year civic disturbance that supplies a relative timeline and helps isolate the 2025 iteration of the biennial event. Knowledge 3 provides the order breakdown at that edition, establishes the criterion "firm orders equal options," and narrows the answer to Riyadh Air. Since these facts are recent and fluid, reliable resolution typically requires external retrieval with search tools.

### E.2 REASONING ACTION SYNTHESIS DATA EXAMPLE

---

**Example: An Overview Reasoning Action Synthesis Data**

**(Clue1: The Location)**: First, the question mentions ..., which presents several potential candidates: the Louvre in Paris with its iconic glass pyramid entrance, .... However, the crucial filtering criterion is the high visitor numbers "during a period of global athletic celebration." This could refer to various major international sporting events: .... Among these candidate cities, only Paris simultaneously possesses both a pyramid-fronted museum and hosted a global athletic celebration, ...., **thus confirming this refers to the Louvre during the 2024 Paris Olympics.**
**(Clue2: The Time)**: Next, the question describes "the year before a citywide nuisance led authorities to convene transit operators," requiring consideration of events in 2023 that could have affected all of Paris and involved the transportation system. Possible candidates include massive public transit strikes, severe air pollution episodes, .... Having established the city as Paris, the "biennial aerospace marketplace named after the city" clearly points to the Paris Air

---

> Show, which is held in odd-numbered years, **thus locking in the timeframe as the 2025 Paris Air Show.**
> **(Clue3: The Order)**: Finally, the question asks which buyer placed "a perfectly balanced commitment with firm orders equal to options." Examining the major order announcements at the 2025 Paris Air Show reveals: .... Therefore, the answer is **Riyadh Air**.

### E.3 AGENT TRAJECTORY EXAMPLE

> **Example: Agent Trajectory**
>
> **Question:** At the biennial aerospace marketplace named after the city whose pyramid-fronted museum recorded high single-digit millions of visitors during a period of global athletic celebration, and where the year before a citywide nuisance led authorities to convene transit operators, which buyer placed a perfectly balanced commitment with firm orders equal to options?
>
> **Step 1**
>
> **<think>**
> ... So we need to identify:
> - A biennial aerospace marketplace named after a city.
> - That city had a pyramid-fronted museum that recorded high single-digit millions of visitors during a period of global athletic celebration (maybe Olympic Games?).
> - "High single-digit millions of visitors" probably means 7-9 million visitors, maybe 8 million. The museum has a pyramid front. Which museum has a pyramid? There is the Louvre Pyramid in Paris: the Louvre Museum has a glass pyramid entrance; it gets about 10 million visitors per year historically (pre-COVID). But high single-digit millions? Actually the Louvre had around 9.5 million visitors in 2019 (maybe earlier?). Also "global athletic celebration" could be Olympics. The 2016 Olympics were in Rio de Janeiro; but Rio's museum? The city might be Rio de Janeiro; it has the Museum of Tomorrow, but not pyramid. Another possibility: The city is Beijing (the National Museum?), the Bird's Nest stadium for Olympics? No.
>
> ...Maybe they refer to the "Louvre Abu Dhabi"? It has a dome, not pyramid. The "Kunsthistorisches Museum"? Not. There's the "Museum of the Iraqi Ministry"? Hmm. Better to search. Let's search web for "pyramid front museum high single digit millions visitors Olympic".
> **</think>**
> **<tool_call>**
> 'query': ['pyramid front museum high single digit millions visitors Olympics', 'pyramid front museum visitors 9 million', 'pyramid front museum visitor numbers Olympic Games', 'pyramid front museum recorded high single-digit millions visitors during Olympics']
> **</tool_call>**
> **<tool_response>**
> A Google search for 'pyramid front museum high single digit millions visitors Olympics' found 10 results:
> Web Results
> 1. [Louvre](https://en.wikipedia.org/wiki/Louvre) It received 8.7 million visitors in 2024, 200,000 less than 2023...
> ...
> A Google search for 'pyramid front museum visitors 9 million' found 10 results:
> Web Results
> 1. [Not just the Louvre: How Europe's museums face the future](https://www.dw.com/en/not-just-the-louvre-how-europes-museums-are-facing-the-future/a-71471539) If you haven't visited the Louvre, then you haven't...
> ...
> **</tool_response>**
>
> **Step 2**
>
> **<think>**
> ...The museum recorded high single-digit millions of visitors during a period of global athletic celebration – likely referring to the Paris Olympics in 2024 ... This refers to some incident in Paris a year before something? "Citywide nuisance" possibly referencing noise or something

that caused public disruption. Let's search for "citywide nuisance led authorities to convene transit operators Paris"
**</think>**
**<tool_call>**
'query': ['citywide nuisance led authorities to convene transit operators Paris year before', 'Paris citywide nuisance transit operators convened', 'Paris citywide nuisance transit operators meeting']
**</tool_call>**
**<tool_response>**
A Google search for 'citywide nuisance led authorities to convene transit operators Paris year before' found 10 results:
Web Results
1. [Innovative solutions for sustainable cities]
(`https://www.ecologie.gouv.fr/sites/default/files/documents/mobility_gb.pdf`) Date published: From long time, France has built up a tightly-meshed mass transit system, upgraded and adapted to new forms of mobility: coverage of mass transit in the...
...
A Google search for 'Paris citywide nuisance transit operators convened' found 10 results:
Web Results
1. [Electric Rental Scooters to be Banned in Paris ...]
(`https://www.nytimes.com/2023/04/03/world/europe/paris-electric-scooters-ban.html`) A referendum emphasized how many residents...
...
**</tool_response>**

| Step 3 |
|---|

**<think>**
The "citywide nuisance" could be referring to the electric scooters, described as a nuisance and ...due to the scooter issue. Let's search for article referencing "authorities convened transit operators" after "citywide nuisance" .
**</think>**
**<tool_call>**
'query': ['"convened transit operators" Paris "electric scooters"', '"convene" "transit operators" Paris "scooter"']
**</tool_call>**
**<tool_response>**
A Google search for '"convened transit operators" Paris "electric scooters"' found 10 results:
Web Results
1. [Rented e-scooters cleared from Paris streets on eve of ban](`https://www.theguardian.com/world/2023/aug/31/rented-e-scooters-cleared-from-paris-streets-on-eve-of-ban`)Paris will become the first European capital to...
...
A Google search for '"convene" "transit operators" Paris "scooter"' found 10 results:
Web Results
1. [U.S. National Blueprint for Transportation Decarbonization ...](`https://www.energy.gov/sites/default/files/2023-01/the-us-national-blueprint-for-transportation-decarbonization.pdf`) Mobile applications can more seamlessly integrate multiple...
...
**</tool_response>**

| Step ... |
|---|

...

| Step 25 |
|---|

**<think>**
...That could be considered perfectly balanced commitment (firm orders equal to options) for Avelo Airlines. But question: "which buyer placed a perfectly balanced commitment with firm orders equal to options?" Let's read that article.
**</think>**

<tool_call>
'url'['https://leehamnews.com/2025/09/10/embraer-e2-finally-break
s-into-the-u-s-market-following-50-firm-orders-50-purchase-rig
hts-from-ulcc-avelo-airlines/'], 'goal': 'Extract details about buyer and number
of firm orders vs options', 'url_info': ['idx': 3, 'query': '"firm orders" "options" "equal" "Paris
Air Show"', 'url': 'https://leehamnews.com/2025/09/10/embraer-e2-final
ly-breaks-into-the-u-s-market-following-50-firm-orders-50-pur
chase-rights-from-ulcc-avelo-airlines/', 'title': 'Embraer E2 Finally Breaks
Into The U.S. Market Following ...', 'snippet': ' options for five more at the 2025 Paris Air
Show, with deliveries beginning in 2028. SAS followed with its largest aircraft order in nearly ',
'hostlogo': None, 'date': '5days ago', 'source': None, 'id': 166]
</tool_call>
<tool_response>
The useful information in https://leehamnews.com/2025/09/10/embraer-e2-f
inally-breaks-into-the-u-s-market-following-50-firm-orders-5
0-purchase-rights-from-ulcc-avelo-airlines/
...Summary: Avelo Airlines, an Ultra-Low-Cost Carrier, has placed a firm order for 50 Embraer
E2-E195 aircraft and has 50 additional purchase rights. The aircraft are scheduled to enter
service with Avelo in mid-2027, with deliveries continuing until 2032.
</tool_response>

| Step ... |
|---|

...

| Step 49 |
|---|

<think>
Not helpful. However we have enough evidence to answer. Thus answer: Riyadh Air. But
perhaps the question expects "Riyadh Air" is the buyer. Let's verify the official naming: "Riyadh
Air" (the new Saudi airline). The order was for 25 Airbus A350-1000s with options for an
additional 25, which is perfectly balanced. So answer: Riyadh Air...
</think>
<tool_call>
'query': ['"ASL Aviation" "firm orders" "options" "Paris Air Show"']
</tool_call>
<tool_response>
A Google search for '"ASL Aviation" "firm orders" "options" "Paris Air Show"' found 10 results:
Web Results
1. [Fleet Renewal - ASL Aviation Holdings](https://www.aslaviationholdings.
com/fleet-renewal/) At ASL Aviation firm orders and 10 purchase rights, announced at
the Paris Air Show in Le Bourget in June 2019...
...
</tool_response>

| Step 50 |
|---|

<think>
... Let's examine context: "the year before a citywide nuisance led authorities to convene transit
operators"... Now, the event where the buyer placed a perfectly balanced commitment with firm
orders equal to options might be at the same air show (Paris Air Show). In 2025, Riyadh Air
placed a perfect balance of 25+25.
Alternatively, in 2019 (previous Paris Air Show), ASL Aviation placed 10+10... So maybe the
question expects us to answer about Riyadh Air because it's the latest.However we need to
consider that "the year before a citywide nuisance led authorities to convene transit operators"
indicates that the nuisance event happened the previous year relative to the statement. Then ask:
which buyer placed a perfectly balanced commitment at the biennial aerospace marketplace?
That suggests the buyer placed that order at the Paris Air Show 2025 (recent). Therefore Riyadh
Air is correct .
Thus answer: Riyadh Air.
We need to provide answer clearly: The buyer is Riyadh Air, which placed a firm order for 25
Airbus A350-1000 aircraft with purchase rights for an additional 25 (equal).
</think>

Thus final answer: Riyadh Air.
**</think>**
**<answer>**
The question references several interconnected events centered around Paris...:
1. **Biennial Aerospace Marketplace**: ...Paris Air Show, officially known as ... attracting manufacturers, airlines, buyers, and aviation professionals from across the world...
2. **Pyramid-Fronted Museum and High Single-Digit Visitor Numbers During Global Athletic Celebration**: ... In 2024, during the period of the Paris Olympic Games—a major global athletic celebration—the Louvre recorded visitor numbers in the high single-digit millions (approximately 8.7 million visitors in 2023 and similar figures in subsequent months)...
3. **Citywide Nuisance and Convening of Transit Operators**: In 2023, Paris experienced a citywide debate on electric scooters (e-scooters), which many residents and officials labeled a "nuisance" due to safety concerns and congestion...
**Recent Developments at the Paris Air Show**
... Among the many deals announced, Riyadh Air stood out for placing a perfectly balanced commitment... This means the number of firm orders equals the number of options, making the commitment symmetrical and strategically flexible.
Key details of Riyadh Air's order:...
**Historical Context and Other Buyers** ... While this demonstrated a similarly balanced commitment, the query appears to focus on recent activity aligning with the context provided (i.e., after the citywide e-scooter nuisance debate and Olympic Games).

| Buyer | Year Announced | Aircraft Model | Firm Orders | Options |
|---|---|---|---|---|
| Riyadh Air | 2025 | Airbus A350-1000 | 25 | 25 |
| ASL Aviation Holdings DAC | 2019 | Boeing 737-800BCF | 10 | 10 |

**Conclusion**
In summary, the buyer that placed a perfectly balanced commitment with firm orders equal to options at the recent Paris Air Show is Riyadh Air...
**<answer>** ✓

