# OpenReview forum: "Scaling Agents via Continual Pre-training"
_ICLR.cc/2026/Conference — ICLR 2026 Poster_

### Official Review · Reviewer_UZu1 · 2025-10-31

**Soundness:** 2
**Presentation:** 2
**Contribution:** 2
**Rating:** 2
**Confidence:** 4

**Summary:**

This paper proposes Agentic Continued Pre-Training for having better base pre-trained models for capabilities requiring tool calls and multi-hop reasoning like DeepResearch. The authors develop a model named AgentFounder-30B using this continued pre-training setup by generating synthetic pre-training data using FAS (First-order Action Synthesis) and HAS (Higher-order Action Synthesis); and further apply SFT on top of the base model with agentic capabilities. They achieve SoTA scores (across open models/deepresearch agents) on various agentic benchmarks like BrowseComp, HLE, GAIA etc.

**Strengths:**

- The paper tackles an important problem of building better base models for agentic capabilities where standard pre-training data is not easily available for multi-hop reasoning, long trajectories, etc.
- The paper's synthetic data generation pipeline is interesting and captures various insights like collecting discarded rejection sampling and historical tool use data, correlation of quality of first-step reasoning with final task completion rates.
- The empirical results are impressive.

**Weaknesses:**

I feel there are quite a lot of points to address before this paper is ready for publication.

- The entire paper reads more like a high-level blog rather than a technical white paper with concrete methodology, steps/details, and experiments.
- The paper tries to present that they are the first to do agentic CPT, but a lot of mid-training/pre-training works do this step before post-training for example: https://arxiv.org/abs/2507.20534
- Lines 65-67 "Consequently, both SFT and RL training depend on limited deterministic
supervisory signals that lock models into replicating specific behavioral patterns rather than develop
flexible decision-making capabilities". The authors don't show any evidence for this and neither do they cite any works highlighting this phenomenon. SFT does mimic specific patterns, but I've not seen this case happening with RL.
- A lot of key technical details are missing (at least in the main paper), like where are the questions sourced from, how many parallel steps are generated for the HAS strategy, what models are used for the rejection sampling, number of tokens for SFT and what's the source of the SFT data, etc.
- How is the model performance affected on non-agentic benchmarks by adding the agentic CPT data? There's no analysis on that.
- What's the effect of the model used for rejection sampling? Can weaker models help as well to eliminate the "distillation" effect of using more capable models to get the rejection sampling data?
- The effect of HAS strategy is minimal on top of FAS, and the evaluation gains seem to be just noise.
- Since I don't know the source of the questions used to generate the synthetic data, how did the authors ensure no contamination with the evaluation sets like HLE.

Although not a weakness, there are a few typos here and there: AgentFoudner instead of AgentFounder in various places, line 224: high-orider instead of high-order.

**Questions:**

I have asked most of my questions in the weaknesses sections above. I am willing to increase my score if the authors are able to address the issues.

---

> ### Author Response · Authors · 2025-11-21
> **Response to Weaknesses 1-4**
>
> **W1: Paper Reads as "Blog vs. Technical Paper"**
> Thank you for your feedback. However, we respectfully emphasize that our work is grounded in **concrete technical contributions**, including the detailed FAS/HAS data synthesis pipeline, a rigorous two‑stage training curriculum, and comprehensive ablation studies (e.g., Tables 4 & 5). Furthermore, we provided in‑depth analysis, with extensive supplementary results included in the Appendix to rigorously validate our methodology.
>
> Simultaneously, we believe that explaining the "why" and sharing our "insights" are crucial contributions. Given the current lack of community consensus on how to best perform agentic mid‑training, we aimed to share our design rationale to benefit the community, rather than just reporting performance metrics.
>
> **\[Revision\]** That said, we recognize the need for greater technical density. In the revised version, we will incorporate granular experimental settings and a more exhaustive description of our methodology to further substantiate the work.
>
> **W2: Novelty of Agentic CPT (vs. Kimi)**
>
> Thank you for raising this. We would like to clarify the distinction:
>
> 1.  The cited work, _Kimi K2_ \[1\], explicitly states that their trajectory data is utilized for **post‑training**, not agenitc CPT/mid-training.
>
> 2.  The GLM 4.5 report \[2\] mentions agentic data in mid‑training but provides **no further details** on methodology or scale, making it irreproducible.
>
> 3.  Our work highlights itself as the first Agentic CPT specifically for **deep research agents.** This is a critical distinction, as these complex, research‑heavy tasks (unlike simpler tool‑use) are where the benefits of CPT are most pronounced and necessary.
>
>
> In summary, the concrete Agentic CPT methodology introduced by AgentFounder remains innovative and highly valuable to the community.
>
> _\[1\] Kimi K2: Open Agentic Intelligence, arXiv 2025._
>
> _\[2\] GLM-4.5: Agentic, Reasoning, and Coding (ARC) Foundation Models, 2025._
>
> **W3: Lines 65-67 (SFT/RL Mimicry Claim)**
>
> We appreciate this correction and acknowledge that our statement regarding RL was imprecise. Our intent was to highlight that SFT and RL primarily drive **convergence** (distribution narrowing) towards specific high‑reward behaviors. Consequently, relying solely on them is often less efficient for exploring the vast action spaces required for agents compared to Agentic CPT.
>
> We will revise this text to accurately reflect this distinction.
>
> **W4: Missing Technical Details**
>
> Thank you for pointing this out. We will incorporate these granular details into the revised paper. Here is a summary of the key specifications:
>
> **\[Question Seed Sources\]** To ensure generalization, we constructed seed questions from diverse streams:
>
> *   _Historical Trajectories__:_ Questions from methods in WebSailor \[3\] and WebShaper \[4\], with trajectories (both positive and negative) from models like WebSailor-72B, WebShaper-72B, QwQ-32B \[5\].
>
> *   _Tool Responses_: Search results (links and snippets) related to over 20 million entities. Furthermore, we matched these with a local build of Common Crawl to simulate the return content for the visit tool.
>
> *   _Documents_: We utilized a local Wikipedia snapshot, Common Crawl data, and millions of public academic papers.
>
>
> **\[HAS Parallel Steps ($N$)\]** In our implementation, the number of alternatives $N$ is sampled from the integer range $[3,5]$.This range was chosen to balance diversity against redundancy and to prevent the total sequence length from exceeding the context window.
>
> **\[Rejection Sampling Model\]** We employed Qwen3‑235B‑A22B‑Instruct‑2507 \[6\] as the judge model. Notably, our internal verification showed high consistency in judgment between this model and other top-tier models like DeepSeek‑V3.1\[7\] and GPT‑4o.
>
> **\[SFT Data Details\]**  We utilized three distinct categories of agentic SFT data. The specific statistics and sources are as follows: (1) **SFT****\‑A:** ~6.4k samples (Avg. ~8.2k tokens). (2) **SFT‑B:** ~14.0k samples (Avg. ~4.7k tokens).  (3) **SFT‑C:** ~8.1k samples (Avg. ~5.9k tokens).
>
> Specifically, the agentic trajectories in both SFT‑A and SFT‑B were constructed following the methodologies of WebSailor \[3\] and WebShaper \[4\], whereas SFT‑C was constructed using the iterResearch \[8\].
>
> _\[3\] WebSailor: Navigating Super-human Reasoning for Web Agent, 2025._
>
> _\[4\] WebShaper: Agentically Data Synthesizing via Information-Seeking Formalization, 2025._
>
> _\[5\] Qwen2.5 Technical Report, 2024._
>
> _\[6\] Qwen3 Technical Report, 2025._
>
> _\[7\] DeepSeek-V3 Technical Report, 2025._
>
> _\[8\] IterResearch: Rethinking Long-Horizon Agents via Markovian State Reconstruction, 2025._

---

> ### Author Response · Authors · 2025-11-21
> **Response to Weakness 5-9**
>
> **W5: Performance on Non-Agentic Benchmarks**
>
> We appreciate this suggestion. To evaluate the impact of Agentic CPT on general capabilities, we tested the model on standard benchmarks including **MMLU**, **SuperGPQA**, and **GPQA**.
>
> | **Benchmark** | **Qwen3-30B-A3B-Base** | **AgentFounder-30B-A3B-Base** |
> | --- | --- | --- |
> | MMLU | 81.38 | 80.11(-1.27) |
> | SuperGPQA | 35.72 | 34.27(-1.45) |
> | GPQA | 43.94 | 42.58(-1.36) |
>
> We consider this marginal regression (averaging ~1.3 points) an acceptable trade-off for the significant gains achieved in agentic capabilities. This marginal drop is primarily attributable to the relatively small proportion of general-purpose data in the mixture compared to the massive volume of agentic data.
>
> **W6: Effect of Rejection Sampling Model**
>
> **\[Reject Sampling Model\]**  In practice, we primarily employed **Qwen3‑235B‑A22B‑Instruct‑2507** for rejection sampling.This choice leverages the **computational asymmetry** between verification and generation: determining data quality (especially with knowledge clues) is significantly easier than generating high‑quality data. Therefore, as shown in **Figure 6**, this allows the model to effectively filter irrational planning action data for Agentic CPT.
>
> **\[On "Distillation" Effect\]** We address the "distillation" concern from two perspectives:
>
> 1.  **Generation:** Our method demonstrates a **weak‑to‑strong** generalization rather than  distillation. Although we utilized only open‑source models for data generation, our final AgentFounder model significantly outperforms them. This proves the pipeline creates net positive capabilities.
>
> 2.  **Filtering:** Can weaker models help? **Yes.** In a pre‑experiment with 1,000 BrowseComp‑like samples, we found that the agreement rate between Qwen3‑235B‑A22B‑Instruct‑2507 and GPT‑4o exceeded **95%**. This high consistency indicates that relatively weaker models (compared to SOTA closed-source models) are sufficient to ensure high data quality.
>
>
> **W7: Effect of HAS Strategy (Minimal/Noise)**
>
> We respectfully clarify that the effect of HAS is significant and robust, particularly in long-context settings.
>
> The HAS data is primarily long‑context (64K‑128K). The ablation in Table 5 was constrained to a **32K** context length due to compute limits, which inherently underrepresents the full benefit of HAS.
>
> The strongest evidence is in **Table 4**. The two‑stage training (where Stage 2 is almost entirely HAS data +10% downsampled Stage 1 data) shows **clear and significant gains** across BrowseComp, BrowseComp‑zh, and GAIA. This demonstrates that HAS provides a substantial boost on top of FAS.
>
> **W8: Data Contamination**
>
> To specifically ensure no contamination with evaluation benchmarks like HLE, we enforced a strict zero‑contamination policy against all test benchmarks using the following strategies:
>
> *   **\[N‑gram Filtering\]** We implemented a strict **13‑gram overlap detection** to exclude any training samples matching the test set. This decontamination strategy aligns with protocols established in **GPT-3 \[9\]**, **PaLM \[10\]**, and **Qwen \[6\]**.
>
> *   **\[URL‑Based Filtering\]** We explicitly excluded domains and URLs known to host evaluation benchmarks (e.g., Hugging Face) to prevent source‑level data leakage.
>
>
> _\[9\] Language models are few-shot learners, in NeurIPS, 2020._
> _\[10\] PaLM: Scaling language modeling with pathways, in JMLR, 2023._
>
> **W9: Typos**
>
> Thank you for your careful reading. We will fix all typos in the final version.
>
>
>
> We sincerely thank you again for the insightful and constructive feedback. We have made our best effort, within our available computational resources, to provide fair and reliable experiments, including the ablation studies. We hope our responses and planned revisions have addressed your concerns, and we would be grateful if you would consider our clarifications in your final evaluation.

---

> > ### Comment · Reviewer_UZu1 · 2025-11-27
> > **Response to Authors**
> >
> > I thank the authors for taking the time to address the concerns and sharing their feedback. Some of my concerns have been alleviated, but some still remain like the distillation effect and rejection sampling analysis. Moreover, the authors don't release the corresponding datasets used for continual PT and SFT, which might be helpful for the community.
> >
> > Assuming the authors add all the additional relevant technical details to the paper and make it more like an ICLR conference paper and also in light of the authors addressing my and other reviewers' concerns, I am increasing my score to 4.

---

### Official Review · Reviewer_cMZ8 · 2025-11-03

**Soundness:** 2
**Presentation:** 2
**Contribution:** 3
**Rating:** 4
**Confidence:** 3

**Summary:**

This paper introduces Agentic Continual Pre-Training as an intermediate training state between pre-training and post-training for building deep research agents. The authors propose a two-stage training pipeline: stage 1 involves standard next-token prediction training on synthetic data, while stage 2 focuses on long-context training. To support the training pipeline, authors propose two key data synthesis methods: First-order Action Synthesis (FAS) for generating planning and reasoning data and Higher-order Action Synthesis (HAS) for transforming wasted trajectories into multi-step multi-choice decision problems.

The trained model achieves strong performance across 10 benchmarks, including BrowseComp, HLE, GAIA, etc. However, the paper does not provide details of some of the claims and doesn't provide enough details about the generated data.

**Strengths:**

- Strong Empirical Performance: The proposed model achieves state-of-the-art results on multiple challenging benchmarks and beats stronger and larger commercial models on most datasets.
- Scalable Data Synthesis: The FAS approach for generating diverse question-answer pairs from entity-anchored knowledge without requiring expensive API calls or human supervision is practical and cost-effective.
- Model Size and Data Scaling: The Authors show that their method scales with the model size and data volume effectively.
- Adaptability to different SFT data: Table 3 shows that the authors' model consistently improves performance across different post-training dataset setups, suggesting the CPT does provide a better foundation.

**Weaknesses:**

- Unverified Claims: The Authors claim that general-purpose foundation models create "inherent optimization conflicts", but this is not verified in this work, and no supporting evidence in the form of a citation is provided. Similarly, authors also claim that "the quality of first-step reasoning exhibits strong positive correlation with final task completion rates". Although this statement may seem intuitive, no experimental validation is provided to support it.
- Unclear distinction from domain adaptation: While the authors claim that the trained model provides a better foundation model for post-training, the paper doesn't verify this claim outside the domain of web-search-based benchmarks. This is understandable because the data was collected using web-search-specific tools; however, the authors should not claim it as a foundation model.
- Limited Transparency on data composition: While the work mentions using 200B and 100B tokens for Stage 1 and 2, respectively, the actual composition of the FAS/HAS data or the data sources is never discussed in the work. How did the authors verify that there was no potential test-data leakage during the training data collection?
- Zero Supervisory Signal: In section 2.2, the authors claim that the data is collected using a zero supervisory signal. However, in section 2.2.2, the authors employ rejection sampling using an LLM-as-a-judge. This is a type of supervision. Authors should clarify this claim.
- Limited Error Analysis: The paper does not provide details error analysis showing what types of tasks/questions benefit most from Agentic CPT versus post-training alone, making it difficult to understand the method's limitations.

**Questions:**

- GLM 4.5 Comparison: Please add some details on how your strategy differs from the GLM 4.5 strategy. What is the difference between the agentic data sizes used for GLM 4.5 and the AgentFounder model?
- FAS vs Trajectory Collection: What is the performance comparison between FAS-generated data and actual collected trajectories (e.g., from rollouts)? Is FAS primarily a cost-saving measure or does it provide qualitative benefits?
- HAS Design Choices: In HAS data construction, you generate N alternatives. What is the value of N here? Please add details about the dataset construction to the paper.
- In the first stage, is there any general-purpose reasoning corpus added to the data mix? If not, does the model suffer from forgetting? Some evaluations on general-purpose language reasoning benchmarks like MMLU would be appreciated and would make the claims stronger.
- Can authors please provide error analysis of the trained model on the evaluation datasets? What type of errors are still prevalent after Agentic CPT? What type of errors are targeted in this training paradigm? This would add significant depth to the work.
- It is not clear to me which data is used for what stage. Some clarification on this would be appreciated.
- In Table 4, what happens if you skip Stage 1 training?
- Evaluation Protocol: The Agentic CPT data is collected using the same set of tools as used in the evaluation. However, I am not sure if the other models were trained to use the same set of tools. This gives an inherent advantage to the AgentFounder model. If my understanding is correct, how does it affect the tool use for AgentFounder vs other models? Are the majority of the errors in baseline models because of incorrect tool use?

---

> ### Author Response · Authors · 2025-11-21
> **Response to Weaknesses 1, 2 and 3**
>
> **W1: Unverified Claims**
>
> Thank you for pointing this out. We address your two points separately.
>
> **\["Inherent Optimization Conflicts"\]** In our work, this term refers to the tension between maintaining diversity in reasoning for broad agentic **exploration** and ensuring convergence towards precise task trajectories (**exploitation**). While we do not provide a quantitative analysis, this phenomenon is well‑documented in related literature, such as the "alignment tax" \[1\] and the exploration‑exploitation trade‑off \[2,3\].
>
> _\[1\] Training language models to follow instructions with human feedback, in Neruips, 2022_.
>
> _\[2\] Soft Actor‑Critic: Off‑Policy Maximum Entropy Deep Reinforcement Learning, In ICML, 2018._
>
> _\[3\] Decoupling Exploration and Exploitation for Meta‑RL, in PMLR, 2021._
>
> **\["First‑Step Reasoning Correlation"\]** We indeed validated the importance of first‑step reasoning in a preliminary study, though details were omitted due to space limits.
>
> *   **\[Experiment\]** We sampled 1,000 **BrowseComp‑like** queries and used solely the first‑step reasoning to predict the final task success.
>
> *   **\[Quantitative Analysis\]** The prediction accuracy reached **80.0%** and **75.2%** for Qwen3‑235B‑A22B‑Thinking‑2507 and Qwen3‑30B‑A3B‑Thinking‑2507, respectively. This empirically confirms a strong correlation between initial planning quality and the final outcome.
>
> *   **\[Qualitative Observation\]** This is further supported by our manual analysis of **error cascading**: agents that initiate an incorrect plan often fail to recover, as they tend to adhere to the flawed initial plan even when subsequent steps provide contradictory information.
>
>
> We will add these experimental details to the Appendix of our revised paper.
>
> **W2: Unclear Distinction from Domain Adaptation**
>
> Thank you for this clarification.
>
> *   **\[Vs. Domain Adaptation\]** We distinguish our work from regular domain adaptation by focusing on learning transferable agentic priors (e.g., planning, reasoning, and decision-making) rather than merely fitting specific data distributions.
>
> *   **\[Agentic Claim\]** Crucially, in this work, we claim an **Agentic** Foundation Model, rather than a **general‑purpose** foundation model.
>
> *   **\[Revision\]** However, we acknowledge that our validation primarily relied on deep research benchmarks. In the revision, we will refine our claims to explicitly frame our work as constructing an **"Agentic Foundation Model for information‑seeking,"** ensuring the scope is strictly aligned with our experimental validation.
>
>
> **W3: Limited Transparency on Data Composition**
>
> We apologize for the lack of clarity. Here is a detailed breakdown of our data composition:
>
> *   **Stage 1 (Agentic CPT, ~200B tokens):** This stage includes ~110B Planning Action data, ~30B Reasoning Action data, and ~40B HAS data. The remaining tokens consist of general-purpose data (_e.g._, mathematics, code, and long‑reasoning) to mitigate catastrophic forgetting of general capabilities.
>
> *   **Stage 2 (Agentic CPT, ~100B tokens):** This stage includes a 10% downsampled portion of Stage 1 data (~20B), ~70B of long‑context HAS data (64K‑128K length), and a small portion of general-purpose tool-use data.
>
>
> **\[On Test-Data Leakage\]** We employed two rigorous strategies to prevent data leakage:
> ● **\[N‑gram Filtering\]** We implemented a strict **13‑gram overlap detection** to exclude any training samples matching the test set. This decontamination strategy aligns with protocols established in GPT-3 \[4\], PaLM \[5\], and Qwen \[6\].
> ● **\[URL‑Based Filtering\]** We explicitly excluded domains and URLs known to host evaluation benchmarks (e.g., Hugging Face) to prevent source-level data leakage.
>
> _\[4\] Language models are few-shot learners, in NeurIPS, 2020._
> _\[5\] PaLM: Scaling language modeling with pathways, in JMLR, 2023._
> _\[6\] Qwen Technical Report, 2023._

---

> ### Author Response · Authors · 2025-11-21
> **Response to Weaknesses 4 and 5**
>
> **W4: "Zero Supervisory Signal" Claim**
>
> Thank you for catching this inconsistency. You are correct that "LLM-as-a‑judge" rejection sampling constitutes a form of supervision.
>
> Our original intent was to emphasize that the FAS data generation requires neither complete trajectory labels (external ground truth) as filtering signals like SFT, nor any human annotation. This fully automated pipeline renders our approach inherently scalable.
>
> We will remove this misleading claim in the revised version to avoid confusion.
>
> **W5: Limited Error Analysis**
>
> We appreciate this insightful suggestion. Based on our analysis, Agentic CPT offers the most significant benefits for tasks involving **tools with high‑dimensional, open‑ended parameter spaces**, such as Web Search (free‑form queries) and Python Code Interpreter.
>
> **\[Insight\]** While SFT effectively learns tool‑call **syntax** (_how_ to call), it often struggles with the **reasoning**required to formulate complex action content (_what_ to search or code). Agentic CPT bridges this gap.
>
> **\[Evidence\]** Table 3 supports this: gains are substantial on **BrowseComp** (search‑intensive) but moderate on **GAIA** (more constrained action space), confirming that our method excels when **the reasoning component** of tool use is the bottleneck.
>
> We will include this analysis in the revised paper.

---

> ### Author Response · Authors · 2025-11-21
> **Response to Questions 1-4**
>
> **Q1: Comparison with GLM 4.5 Strategy**
>
> The GLM 4.5 technical report \[7\] does not disclose details regarding its agentic data synthesis methodology or the scale of agentic trajectories. Consequently, a direct quantitative comparison of data strategies is not feasible.
>
> However, this highlights a key contribution of our work: we propose a **concrete, low‑cost, and scalable method** for creating massive agentic CPT datasets, bridging a gap in the current literature with a transparent "how-to" framework.
>
> **\[Experiments\]** Furthermore, our results demonstrate a significant performance advantage: AgentFounder surpasses GLM 4.5 by **margins of +13.5%, +5.8%, and +6.8%** on BrowseComp, BrowseComp-zh, and GAIA, respectively.
>
> _\[7\] GLM‑4.5: Agentic, Reasoning, and Coding (ARC) Foundation Models, 2025._
>
> **Q2: FAS vs. Full Trajectory Collection**
>
> We appreciate this question. While cost‑efficiency is a core contribution of FAS, our empirical results demonstrate that it also offers superior **learning efficiency** compared to full trajectories.
>
> **\[Performance Comparison\]** We conducted a controlled experiment using a fixed budget of 15B training tokens on top of the SFT-A baseline. The results on BrowseComp are as follows:
>
> | CPT Data Source | None (Baseline) | **Full Trajectories** | FAS Data（ours） |
> | --- | --- | --- | --- |
> | Pass@1 | 26.9% | 27.1%(**+2.3%**) | 29.2%(**+0.2%**) |
>
> These results indicate that FAS significantly outperforms full trajectories under the same compute budget.
>
> **\[Qualitative Benefits\]** We attribute this efficiency to three structural advantages:
>
> **Higher Data Density.** FAS samples (containing only 1 interaction) are significantly shorter than full trajectories (containing multiple interactions). This allows the model to encounter a much wider variety of seed problems within the same token budget.
>
> **High Signal‑to‑Noise Ratio.** Full trajectories are often dominated by environmental information (e.g., search results, webpages). FAS strips this away to focus computation strictly on the core planning logic.
>
> **Production Efficiency.** FAS generation bypasses the latency and QPS bottlenecks of external environments. This makes the data synthesis pipeline substantially more robust and scalable.
>
> **Q3: HAS Design Choices (Value of $N$)**
>
> Thank you for requesting this detail. In our implementation, the number of alternatives $N$ is sampled from the integer range $[3,5]$. We identified this range as an optimal trade-off for two main reasons:
>
> *   **\[Semantic Redundancy\]** If $N$ is too large, the generated alternatives tend to become semantically indistinguishable (i.e., too similar), leading to diminishing returns in learning efficiency.
>
> *   **\[Context Overflow\]**  An excessive number of alternatives increases the risk of exceeding the model's maximum context window (e.g., 128K).
>
>
> We will explicitly specify these hyperparameters in the revised paper.
>
> **Q4: General-Purpose Reasoning and Forgetting**
>
> Yes, as detailed in the response to  **W3**, we incorporated general‑purpose data into our mix to mitigate forgetting. To evaluate the impact, we ran experiments on general benchmarks including MMLU, SuperGPQA, and GPQA. The results are as follows:
>
> | **Benchmark** | **Qwen3-30B-A3B-Base** | **AgentFounder-30B-A3B-Base** |
> | --- | --- | --- |
> | MMLU | 81.38 | 80.11(-1.27) |
> | SuperGPQA | 35.72 | 34.27(-1.45) |
> | GPQA | 43.94 | 42.58(-1.36) |
>
> We consider this slight degradation an acceptable trade‑off for the significant gains in agentic capabilities. This marginal drop is primarily attributable to the relatively small proportion of general‑purpose data in the mixture compared to the massive volume of agentic data.

---

> ### Author Response · Authors · 2025-11-21
> **Response to Questions 5-8**
>
> **Q5: Error Analysis of AgentFounder**
>
> We appreciate this suggestion. We break down the error analysis into persistent limitations and targeted improvements.
>
> **\[Persistent Limitation: Domain Knowledge\]** AgentFounder enhances **how** to solve problems (reasoning/planning), but not **what** the model knows (domain facts). While search/visit tools expand knowledge boundaries, the effective direction of a search still relies on the model's internal parametric knowledge, which Agentic CPT does not explicitly expand.
>
> **\[Targeted Improvements\]** Our analysis reveals that Agentic CPT introduces specific behavioral patterns that **reduce planning errors** and **enhance error recovery potential**. We detail these improvements in two aspects:
>
> *   **\[Planning Quality\]** We sampled 1,000 unseen BrowseComp‑like queries. To ensure rigorous evaluation, we assessed the quality of the first‑step plan using constructed reference knowledge and carefully designed rubrics.
>
>
> | **Model** | **High-Quality Plan** | **Pass@1 Gain** | **Pass@3 Gain** |
> | --- | --- | --- | --- |
> | Qwen3-30B-A3B | 44.1% | \- | \- |
> | AgentFounder-30B-A3B | **76.6% (+32.5%)** | **+17.3%** | **+25.2%** |
>
> AgentFounder achieves a massive gain in planning quality (+32.5%). Notably, this result aligns well with the improvements in end-to-end accuracy (Pass@1 +17.3% and Pass@3 +25.2%).
>
> *   **\[Error Resilience\]** We analyzed 100 trajectories involving tool failures (e.g., page access errors). We employed Qwen3‑235B‑A22B‑Thinking‑2507 to determine whether the agent abandoned the current exploration direction, using the subsequent ReAct content as input for evaluation.
>
>
> | **Metric** | **Qwen3-30B-A3B** | AgentFounder-30B-A3B |
> | --- | --- | --- |
> | **Give-up Rate** | 81.0% | **65.0%（-16.0%）** |
>
> The results show that AgentFounder significantly reduces the Give‑up Rate, effectively mitigating the agent's tendency to abandon failed paths.  For example, the agent actively attempts to access mirror sites to recover from failures, rather than simply abandoning the direction.
>
> **Q6: Data-to-Stage Mapping**
>
> We have provided a detailed breakdown of the specific data composition for each training stage in our response to **W3**. Please kindly refer to that section for the full details.
>
> **Q7: Ablation Study (Skipping Stage 1)**
>
> We appreciate this suggestion. In AgentFounder, the core distinction between Stage 1 and Stage 2 is **context length**. The primary objective of Stage 2 is to adapt the model to extended context windows, a "short‑to‑long" curriculum strategy widely recognized by the community (e.g., GLM-4.5 and Qwen).
>
> While we could not perform an ablation training only Stage 2 due to **prohibitive computational costs**, the substantial performance gains demonstrated in Table 4 sufficiently validate the rationality and effectiveness of our two-stage training strategy.
>
> **Q8: Fairness of Evaluation Protocol**
>
> We would like to clarify this point, as we believe the comparison is fair:
>
> 1.  **Tool Similarity:** The tools used in web benchmarks are highly standardized. The Search tool typically wraps the Google Search API, Visit uses a standard parser like Jina Reader, and the Python interpreter is largely identical. Any minor implementation differences would have a negligible impact on performance.
>
> 2.  **Baseline Scores:** To ensure fairness, we **prioritized using the officially reported scores** from the baseline models' respective papers. In those cases, the models were trained and evaluated using their own consistent set of tools.
>
>
> Therefore, the significant performance gains of AgentFounder do not stem from a simple advantage in tool adaptation, but from a genuine improvement in the underlying reasoning and planning capabilities.
>
> We sincerely thank you again for your insightful and constructive feedback. We have made our best effort, within our computational constraints, to provide fair and rigorous validation, including the additional ablation studies. We hope our responses have fully addressed your concerns, and we would be grateful if you could **kindly reconsider your assessment** in light of these clarifications and improvements.

---

> > ### Comment · Reviewer_cMZ8 · 2025-11-26
> > **Response to Authors**
> >
> > Thank you for providing a detailed rebuttal addressing all the questions. In light of the clarifications and additional analysis presented by the authors, I find that if these results are added to the paper, it will immensely benefit the clarity and soundness of the work.
> > The authors have done a commendable job with this work, and thus, I have increased my score and confidence accordingly.

---

### Official Review · Reviewer_pAr1 · 2025-11-08

**Soundness:** 3
**Presentation:** 3
**Contribution:** 3
**Rating:** 8
**Confidence:** 4

**Summary:**

This paper proposes Agentic Continual Pre-Training, an intermediate training stage between pre-training and post-training to bridge the optimization conflict between pre-training and agent post-training. To test the effectiveness of the proposed method, the authors further synthesized large-scale data for model training. The resulting model, AgentFounder-30B, achieves state-of-the-art results on 10 benchmarks, including 39.9% on BrowseComp-en, 31.5% on HLE, and 72.8% on GAIA. Ablations show consistent improvements, and scaling analyses indicate steady gains with larger models and data. Key contributions include:

(1) First-order Action Synthesis (FAS): reformulates static knowledge into complex questions and synthesizes planning/reasoning behaviors without API costs.

(2) Higher-order Action Synthesis (HAS): reuses suboptimal trajectories by expanding each step into contrastive multi-choice decision processes.

(3) Two-stage CPT strategy: progressive training with 32K and 128K contexts (200B + 100B tokens) to enhance long-horizon reasoning.

**Strengths:**

(1) The data synthesis pipeline is scalable and efficient.  FAS enables large-scale agentic data generation through knowledge-to-question transformation without costly API calls, while HAS reuses suboptimal trajectories by converting them into step-level decision processes. Together, they improve data efficiency and coverage. The addition of reject sampling with knowledge alignment verification further enhances data quality.

(2) The experiment results are promising. AgentFounder-30B achieves SOTA across ten benchmarks, outperforming all open-source agents and even surpassing some commercial systems. It is the first open-source model to exceed 30% on HLE (31.5%), with consistent gains across all SFT configurations, demonstrating strong robustness.

**Weaknesses:**

(1) Limited Theoretical Evidence for “Optimization Conflict”
The claimed conflict between capability learning and alignment is intuitively appealing but empirically unsupported. It would be great if the authors could provide more evidence for this claim, like gradient interference analyses (e.g., cosine similarity between CPT and SFT objectives), visualize loss landscapes, and probe capability retention during SFT with vs. without CPT.

(2) Lack of analysis regarding the influence of the model on data synthesis.  It would be great if the authors could provide more details on the models used in data synthesis. Ideally, it would be great if the authors could further provide an oblation study regarding model performance against the quality of the model for data synthesis.

**Questions:**

The paper's core contribution relies heavily on the synthetic FAS and HAS data synthesis. Could you specify the data release plan, like whether the whole CPT stage training data will be released? If it's not, it's possible to release a sample subset for community inspection.

---

> ### Author Response · Authors · 2025-11-21
>
> **W1. Theoretical Evidence for “Optimization Conflict”**
>
> We appreciate the your recognition that this concept is intuitively appealing. In our work, this "Optimization Conflict" refers to the tension between maintaining **diversity** in reasoning for broad agentic exploration and ensuring **convergence** towards precise task trajectories (exploitation). While we do not provide a quantitative analysis (e.g., gradient interference), this phenomenon is well-documented in related literature, such as the "alignment tax" \[1\] and the exploration-exploitation trade-off \[2,3\].
>
> _\[1\] Training language models to follow instructions with human feedback, NeurIPS 2022._
>
> _\[2\] Soft Actor-Critic: Off-Policy Maximum Entropy Deep Reinforcement Learning, ICML 2018._
>
> _\[3\] Decoupling Exploration and Exploitation for Meta-RL, PMLR 2021._
>
> **W2.** **Model Selection on Data Synthesis**
>
> We appreciate this valuable suggestion. We agree that the choice of the synthesizer model is critical to data quality. While a full-scale quantitative ablation across multiple large models was **computationally prohibitive**, given the combined constraints of the **massive scale of our synthesis pipeline** and the **expensive training cost** for each ablation run, we provide detailed specifications of the models used and the empirical pre-experiments that drove our selection.
>
> We primarily utilized **Qwen3‑235B‑A22B‑Thinking‑2507** (referred to as Qwen3‑235B) as our core synthesizer, supplemented by **DeepSeek‑R1** and **DeepSeek‑V3.1** to ensure diversity. All utilized models are open-source.
>
> **\[Observations\]** We compared Qwen 2.5‑72B against Qwen3‑235B‑A22B‑Thinking‑2507 during our pre‑experiments. We observed two key distinctions that drove our selection:
>
> 1.  **Question Synthesis.** While both models generate factually correct QA pairs, Qwen3‑235B excels at constructing high‑difficulty, multi‑hop queries that utilize intricate logical relationships between entities. In contrast, Qwen 2.5‑72B tends to be conservative, often merely listing conditions **without building deep logical structures.**
>
> 2.  **Action Generation.** Qwen3‑235B produces trajectories with higher reasoning densityand explicit intermediate steps. Qwen 2.5‑72B, while superficially coherent (using logical connectors), often exhibits **logical leaps** where critical reasoning links are missing.
>
> Consequently, we selected Qwen3‑235B‑A22B‑Thinking‑2507 as the primary synthesizer to ensure data quality.
>
> While we acknowledge that employing even stronger models could potentially further enhance learning efficiency, AgentFounder serves as a compelling proof‑of‑concept for **weak‑to‑strong capability amplification**. This is evidenced by the fact that our trained AgentFounder-30B-A3B significantly outperforms all the open-source models utilized in its data synthesis.
>
> **Q1. Data Release**
>
> We appreciate your interest in our data assets. While the sheer volume and data compliance regulations prevent us from releasing the complete agentic CPT corpus, we remain committed to contributing to the open‑source community.
>
> We plan to release a high‑quality, representative sample subset of the FAS and HAS data upon publication. This release will cover various action types and difficulty levels, enabling the community to better understand our synthesis pipeline and verify our data quality.
>
>
> We sincerely thank you again for your insightful and constructive feedback on our work. We hope these clarifications and improvements further strengthen the paper. If you have any remaining questions or require further details, we would be very happy to provide them.

---

### Author Response · Authors · 2025-12-01
**Overview of the Rebuttal**

We sincerely thank all reviewers for their time and constructive feedback. We are encouraged by their positive recognition of our work's data scalability, solid empirical results, and research significance.

Here, we succinctly summarize the strengths highlighted by the reviewers and provide a roadmap of our key clarifications regarding common concerns.

### Strengths

- **Methodology & Data Synthesis**
    - **Scalable and Cost-effective**: "The data synthesis pipeline is scalable and efficient." **(pAr1)**; "The FAS approach ... without requiring expensive API calls or human supervision is practical and cost-effective." **(cMZ8)**
    - **Insightful Pipeline Design**: "The paper's synthetic data generation pipeline is interesting and captures various insights..." **(UZu1)**
    - **Quality Enhancement**: "The addition of reject sampling with knowledge alignment verification further enhances data quality." **(pAr1)**

- **Evaluation & Empirical Results**
    - **SOTA & Commercial Comparison**: "AgentFounder-30B achieves SOTA across ten benchmarks..." **(pAr1)**; "beats stronger and larger commercial models on most datasets." **(cMZ8)**
    - **Impressive Outcomes**: "The empirical results are impressive." **(UZu1)**; "The experiment results are promising." **(pAr1)**

- **Significance, Scalability & Robustness**
    - **Addressing Critical Problems**: "The paper tackles an important problem of building better base models for agentic capabilities where standard pre-training data is not easily available for..." **(UZu1)**
    - **Effective Scaling**: "The Authors show that their method scales with the model size and data volume effectively." **(cMZ8)**
    - **Adaptability & Robustness**: "model consistently improves performance across different post-training dataset setups, suggesting the CPT does provide a better foundation." **(cMZ8)**; "Consistent gains across all SFT configurations, demonstrating strong robustness." **(pAr1)**

### Common Concerns and Rebuttal Highlights

- **C1. Data Composition and Details _(cMZ8 W3/Q6; UZu1 W4)_**
    - In our reply to Reviewer cMZ8 (W3), we have **detailed the data composition** for each Agentic CPT stage, including the breakdown of FAS, HAS, and general data. Furthermore, we have clarified the sources of seed questions and SFT data to address the points raised by Reviewer UZu1 (W5).

- **C2. General/Non-agentic Benchmarks _(cMZ8 Q4; UZu1 W5)_**
    - To demonstrate that AgentFounder maintains strong general capabilities, we have provided experimental results for AgentFounder-30B on **three general benchmarks (MMLU, SuperGPQA, and GPQA)** in our replies to Reviewer cMZ8 (Q4) and UZu1 (W5).

- **C3. Training Pipeline Comparison with GLM4.5/Kimi-K2 _(cMZ8 Q1; UZu1 W2)_**
    - **Addressed** in the reply to cMZ8 (Q1): We noted that while GLM4.5 does **NOT disclose details** regarding its agentic trajectory training data, and our model achieves **higher scores on Browsecomp-en, Browsecomp-zh and GAIA benchmarks**.
    - Clarified for UZu1 (W2): We pointed out that Kimi explicitly states its trajectory data is used for post-training, rather than pre-training or mid-training, to avoid any misunderstanding.

- **C4. How to avoid Data Contamination _(cMZ8 W3; UZu1 W8)_**
    - We have outlined our two mechanisms for avoiding potential data contamination: **N‑gram Filtering** and **URL‑Based Filtering**.

- **C5. The Number of HAS Choices _(cMZ8 Q3)_**
    - We specified that $N$ is set to $[3, 5]$ and justified this choice based on constraints regarding **semantic diversity** and **context window size**.

- **C6. Reject Sampling Model Selection _(pAr1 W2; UZu1 W4 & W6)_**
    - We identified the open-source models used for data generation (e.g., Qwen3 series, DeepSeek series) and the judge model used for rejection sampling.
    - Additionally, we discussed the insight regarding computational asymmetry (for UZu1 W6) and shared our observations on model selection (for pAr1 W2).

- **C7. Statement on "Optimization Conflict" _(pAr1 W1; cMZ8 W1)_**
    - We have elaborated on the specific meaning of "optimization conflict" in this context and cited prior research to support the validity of this claim. We appreciate that Reviewer pAr1's technical intuition aligns with this perspective.

- **C8. FAS vs. Full Trajectory Collection _(cMZ8 Q2)_**
    - We presented a quantitative experimental analysis in response to cMZ8 (Q2). We also summarized qualitative arguments regarding data density, high signal-to-noise ratio, and production efficiency to demonstrate why FAS data is more suitable for Agentic CPT.

- **C9. Data Release _(pAr1 Q1; UZu1's reply)_**
    - Following the suggestions of Reviewers pAr1 and UZu1, we will release a representative subset of FAS and HAS data samples upon acceptance, enabling the community to better understand our synthesis pipeline and verify our data quality.

**Note**: Please refer to the individual responses for detailed evidence and experiments.

---

### Author Response · Authors · 2025-12-01
**Rebuttal Outcome Summary: Scores Improved to 8, 8, 4**

Dear Area Chair, Senior Area Chairs, and Program Chairs,

We sincerely appreciate the reviewers' positive recognition of our rebuttal, leading to improved scores of **8, 8, 4**.

---

Below is a summary of the post-rebuttal updates and evidence:

*   **Reviewer pAr1 (Score: 8):** Maintained strong support with a score of **8**.
*   **Reviewer cMZ8 (Score: $\uparrow$ 8):** Fully acknowledged our response and increased the score to **8** on **Nov 26** (prior to the openreview data leak).
    *   Comment: *"The authors have done a commendable job with this work, and thus, I have **increased my score and confidence** accordingly."*
    *   Link: [Evidence Record (OpenReview History)](https://openreview.net/revisions?id=6a0JRD6YQr)
*   **Reviewer UZu1 (Score: $\uparrow$ 4):** Acknowledged our rebuttal and raised the score consistent with their initial comment.

    *   Initial Comment: *"I am willing to **increase my score** if the authors are able to address the issues."*
    *   Comment: *"in light of the authors addressing my and other reviewers' concerns, I am **increasing** my score to 4."*
    *   Link: [Evidence Record (OpenReview History)](https://openreview.net/revisions?id=d09lQmo3tZ)

We deeply recognize the immense workload involved in your role. We respectfully highlight that our rebuttal has been well-received by the reviewers, and we are committed to further refining the details of our work.

If you have any remaining questions, we are more than happy to address them.

---

### Meta-Review · Area_Chair_uw2R · 2026-01-17

**Summary:**

The paper received mixed reviews from reviewers. The main points raised were:

1. Theoretical claims about "optimization conflict" (3 reviewers)

2. Lack of technical detail regarding the models for synthesis, hyper-parameters, and composition of the data (3 reviewers)

3. Catastrophic forgetting, data contamination, and zeor-supervision claims

4. Comparison to related work

**Reviewer Concerns:**

The authors did a good job responding in the rebuttal.

1. They clarified that optimization conflict refers to the classic exploration / exploitation trade-off, and they agreed to revise the paper.

2. They provided more details about the models, data, and hyper-parameters.

3. They presented new results about catastrophic forgetting, and they also agreed to revise the paper regarding the zero supervision claim.

4. They clarified the distinction to related work.

Overall, they addressed each of the points.

**Reviewer Scores:**

The reviewers were initially mixed about the paper, but the authors did a good job at responding to each of the points. They agreed to revise the paper, which lead to a stronger submission. I believe the reviewers would have appreciated these points and supported accepting the paper.

---

### Decision · Program_Chairs · 2026-01-26

Accept (Poster)